# Mechanism of action of non-camptothecin inhibitor Genz-644282 in topoisomerase I inhibition

Masahiro Nishida[1,6], Takeshi Terabayashi [2,6], Shigeru Matsuoka[3], Tomoko Okuma[4], Sawako Adachi[4], Tadashi Tomo[1], Masanori Kawano[5], Kazuhiro Tanaka[5], Hiroshi Tsumura[5], Hirofumi Anai[1], Toshimasa Ishizaki[2], Yoshihiro Nishida[4 ✉] & Katsuhiro Hanada [1 ✉]

Topoisomerase I (TOP1) controls the topological state of DNA during DNA replication, and its dysfunction due to treatment with an inhibitor, such as camptothecin (CPT), causes replication arrest and cell death. Although CPT has excellent cytotoxicity, it has the disadvantage of instability under physiological conditions. Therefore, new types of TOP1 inhibitor have attracted particular attention. Here, we characterised the effect of a non-camptothecin inhibitor, Genz-644282 (Genz). First, we found that treatment with Genz showed cytotoxicity by introducing double-strand breaks (DSBs), which was suppressed by co-treatment with aphidicolin. Genz-induced DSB formation required the functions of TOP1. Next, we explored the advantages of Genz over CPT and found it was effective against CPT-resistant TOP1 carrying either N722S or N722A mutation. The effect of Genz was also confirmed at the cellular level using a CPT-resistant cell line carrying N722S mutation in the *TOP1* gene. Moreover, we found arginine residue 364 plays a crucial role for the binding of Genz. Because tyrosine residue 723 is the active centre for DNA cleavage and re-ligation by TOP1, asparagine residue 722 plays crucial roles in the accessibility of the drug. Here, we discuss the mechanism of action of Genz on TOP1 inhibition.

[1] Clinical Engineering Research Centre, Oita University, Yufu, Oita, Japan. [2] Department of Pharmacology, Oita University, Yufu, Oita, Japan. [3] Department of Clinical Pharmacology & Therapeutics, Oita University, Yufu, Oita, Japan. [4] Department of Obstetrics and Gynecology, Oita University, Yufu, Oita, Japan. [5] Department of Orthopaedic Surgery, Faculty of Medicine, Oita University, Yufu, Oita, Japan. [6]These authors contributed equally: Masahiro Nishida, Takeshi Terabayashi. ✉email: ynishida@oita-u.ac.jp; hanada@oita-u.ac.jp

The catalytic activity of TOP1 involves cleavage of one of the two strands of double-stranded DNA by the hydroxy group of its tyrosine residue 723 and linking of the phosphate group at the 3'-end of the strand to form a binary DNA–TOP1 covalent complex[1,2]. After cleavage, the DNA strand with a single-strand break can rotate around the unbroken strand to remove DNA supercoils. Subsequently, the cleaved strand is re-ligated by TOP1. Thus, TOP1 controls the topological state of DNA. Because supercoiled DNA typically appears in front of replication forks during DNA replication, a number of studies have shown that the dysfunction of TOP1 due to treatment with camptothecin (CPT) causes stalled replication forks[3–5]. These stalled replication forks are often converted into double-strand breaks (DSBs) of DNA by the action of MUS81–EME1/2 structure-specific endonuclease[4,6]. A characteristic feature of DSB formation induced by a TOP1 inhibitor is that DSBs are completely suppressed by co-treatment with a DNA replication inhibitor such as aphidicolin[6,7]. CPT also induces DSBs outside of S phase by a transcription-dependent mechanism, and is also cytotoxic in non-proliferating cells[8]. Because the cytotoxic effect of CPT mostly appears in proliferating cells, CPT derivatives have been used in a clinical context as anticancer medicines. However, several limitations of CPT were also identified. First, it is unstable under physiological conditions. Although the hydroxylactone ring of CPT, also called the E-ring, is important for CPT activity, it is easily hydrolysed and converted into the carboxylate form, which causes defective TOP1 inhibition[9,10]. Second, CPT is easily removed from cells by ABC transporters such as ABCG2[11]. Finally, various CPT-resistant mutations have been discovered in the *TOP1* gene. W736X and G737S mutations were discovered in patients with non-small cell lung cancer treated with irinotecan[12]. In addition, various CPT-resistant mutations in the *TOP1* gene have also been generated in tissue culture studies, such as F361S[13], G363C[14,15], R364H[16], E418K[17,18], G503S[18–20], D533G[18–20], L617I[21], R621H[21], A653P[15], E710G[21,22], G717V[23], and N722S[19,20]. Using these mutants, the action of CPT has been investigated. Among the residues of TOP1, arginine at position 364, aspartate at 533, and asparagine at 722 play crucial roles in the direct interaction between CPT and TOP1. Therefore, impaired interaction with CPT caused by R364H, D533G, and N722S mutations was found to be associated with CPT resistance[24]. F361S, G363C, G503S, L617I, R621H, A653P, E710G, and G717V mutations change the architecture of the active site, which results in reduced affinity to CPT[15,21,25,26]. The interaction between asparagine at position 722 of TOP1 and CPT contributes to CPT accessing its active site, the tyrosine residue at position 723. However, the mechanism by which these mutations cause CPT resistance remains incompletely understood.

Two approaches have been taken to overcome the limitations of CPT. One is to modify it, especially its E-ring, to improve its activity and stability. Accordingly, many CPT derivatives have been generated and used in clinical and biological studies[27]. The other involves screening novel compounds that act against TOP1. Nitidine, a natural product classified as a benzo[c]phenanthridine alkaloid, was discovered as a candidate TOP1 inhibitor with good antitumor potency[28]. To improve the inhibitory effect of nitidine on TOP1, a synthetic compound of this class, Genz-644282 (Genz) [8,9-dimethoxy-5-(2-N-methylaminoethyl)-2,3-methylenedioxy-5H-dibenzo[c,h][1,6]naphthyridin-6-one], was designed (Fig. 1a)[29–31]. After treatment with Genz, the accumulation of DNA–TOP1 complex was detected by the in vivo complex of enzyme (ICE) assay, and the formation of γ-H2AX foci was observed to be induced[29]. These cytotoxic effects of Genz were quite similar to those of CPT. The advantage of Genz involved exhibiting cytotoxicity against a CPT-resistant cell line that carries the N722S mutation, while no advantage was found on another CPT-resistant cell line that carries R364H mutation[29]. However, little is known about the pharmacological effects of Genz.

In this study, we characterised the pharmacological effects of Genz from two perspectives: one involving the molecular mechanism by which Genz exhibits cytotoxicity in comparison to CPT and the other focusing on the advantages of Genz over CPT. Through these investigations, we found that the arginine 364th residue of TOP1 played a crucial role for Genz as well as CPT. The different binding of Genz from CPT on TOP1 provides a possible mechanism explaining how Genz acts on CPT-resistant TOP1 carrying a mutation at its 722nd residue. Here, we discuss how Genz acts to CPT-resistant TOP1 carrying N722A or N722S mutation.

## Results

**Investigation of DSB formation induced by Genz-644282 treatment.** To understand the cytotoxic activity of Genz, DSB formation after treatment with it was investigated. First, cells were treated with Genz, and the formation of γ-H2AX foci and redistribution of 53BP1 foci were analysed. Under untreated conditions, γ-H2AX foci were rarely observed, while after Genz treatment, cells with γ-H2AX foci appeared from 1 h and cells positive for γ-H2AX foci were increased by 16 h of treatment (Supplementary Fig. 1a, c). In contrast, 53BP1 showed spontaneous nuclear speckles (large aggregated foci) in the SV40-transformed MRC5 cell line[32]. When DSBs were introduced by DNA damaging agents, 53BP1 was redistributed to the broken DNA ends and formed small but bright foci[32–34]. Under untreated conditions, 53BP1 showed nuclear speckles, while after Genz treatment, the damage-induced formation of 53BP1 foci started to occur from 8 h of treatment and increased by 24 h of treatment (Supplementary Fig. 1a, d). The dynamics of the formation of γ-H2AX and 53BP1 foci after treatment with Genz were quite similar to those with CPT (Supplementary Fig. 1b–d). The formation of γ-H2AX foci represents not only DSB formation but also single-strand breaks and gaps caused by stalled DNA replication, whereas the redistribution of 53BP1 foci occurs only after the appearance of DSBs. Because cells positive for γ-H2AX foci started to accumulate earlier than those with 53BP1 foci (Supplementary Fig. 1), it suggested the possibility that stalled DNA replication forks are initially induced by treatment with Genz, and these stalled forks are subsequently converted to DSBs. To address this possibility, we first investigated whether Genz-induced DSB formation occurred at DNA replication sites. DNA replication sites were pulse-labelled with EdU for 15 min and then Genz and CPT treatments were applied. After 2 h, cells were fixed and stained with EdU and γ-H2AX, after which we analysed whether EdU foci and γ-H2AX foci were colocalised. Because EdU foci were overlapped with γ-H2AX foci, Genz-induced DSB formation would occur around DNA replication sites (Fig. 1b). This phenomenon is quite similar to the findings for CPT[4].

To explore the mechanism behind Genz-induced DSB formation, we have used PFGE to assess whether combined treatment with a DNA replication inhibitor, aphidicolin, could suppress the DSB formation due to Genz because previous studies showed that DSBs induced by topoisomerase I inhibitors, including CPT, do not accumulate after combined treatment with aphidicolin[6,7]. DSBs represented as 'broken DNA' and 'chromosome fragmentation in apoptosis' were suppressed by combined treatment with aphidicolin (Fig. 1c–e). To rule out the possibility that Genz-induced DSBs were produced as a result of apoptosis, we also examined Genz-induced DSB formation in the presence of the apoptosis inhibitor Z-VAD-FMK[35]. As expected, the accumulation of 'broken DNA' representing Genz-induced DSBs was

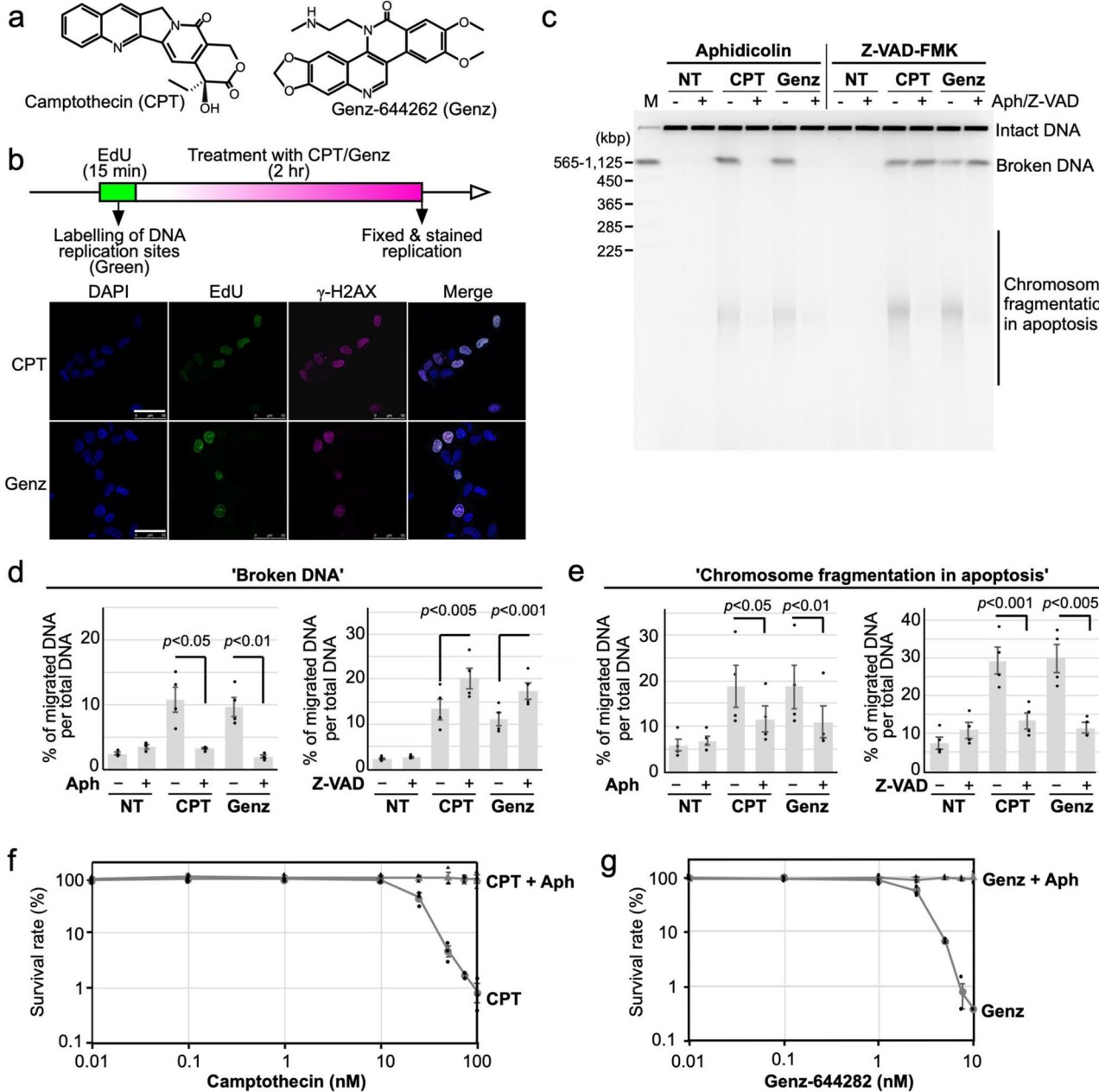

**Fig. 1 Analysis of the cytotoxic effects of CPT and Genz. a** Chemical structures of camptothecin (CPT) and Genz-644282 (Genz). **b** Immunofluorescence analysis of the formation of EdU and γ-H2AX foci. Cells were pulse-labelled with 10 μM EdU for 15 min. Subsequently, these pulse-labelled cells were treated with 1 μM CPT or 1 μM Genz, and the formation of EdU and γ-H2AX foci was analysed. The scale bars represent 50 μm. **c** PFGE analysis of DSB accumulation after 24 h-treatment with a DNA replication inhibitor, aphidicolin (10 μM), combined with 1 μM CPT and 1 μM Genz, and PFGE analysis of DSB accumulation after 24 h-treatment with an apoptosis inhibitor, Z-VAD-FMK (10 μM), combined with 1 μM CPT and 1 μM Genz. **d** Quantification of CPT- and Genz-induced 'broken DNA', which represents DSB formation due to DNA stress. The data are presented as the percentages of the amount of broken DNA relative to total DNA (intact DNA + broken DNA + chromosome fragmentation in apoptosis). **e** Quantification of 'chromosome fragmentation in apoptosis' induced by treatments with CPT and Genz. The data are presented as the percentages of the amount of chromosome fragmentation in apoptosis per total DNA (intact DNA + broken DNA + chromosome fragmentation in apoptosis). The means and standard deviations were determined from four independent experiments. **f** Survival curve of SV40-transformed human fibroblasts, MRC5 against CPT. Indicated concentrations of CPT were applied for 24 h. Circle dots represents results of survival rates after treatment with CPT. Triangle dots represents results of survival rates after combined treatment with CPT and 10 μM aphidicolin. **g** Survival curve of CPT on SV40-transformed MRC5 against Genz. Indicated concentrations of CPT was applied for 24 h. Circle dots represents results of survival rates after treatment with CPT. Triangle dots represents results of survival rates after combined treatment with CPT and 10 μM aphidicolin.

increased, whereas 'chromosome fragmentation in apoptosis' was suppressed by combined treatment with Z-VAD-FMK (Fig. 1c±e). These results indicate that the treatment with Genz induced DSBs around DNA replication sites rather than chromosome

fragmentation in apoptosis. Previously, it was reported that the cytotoxicity of CPT was suppressed by combined treatment with aphidicolin[7]. We also confirmed this in the current study (Fig. 1f). Therefore, we assessed whether combined treatment with

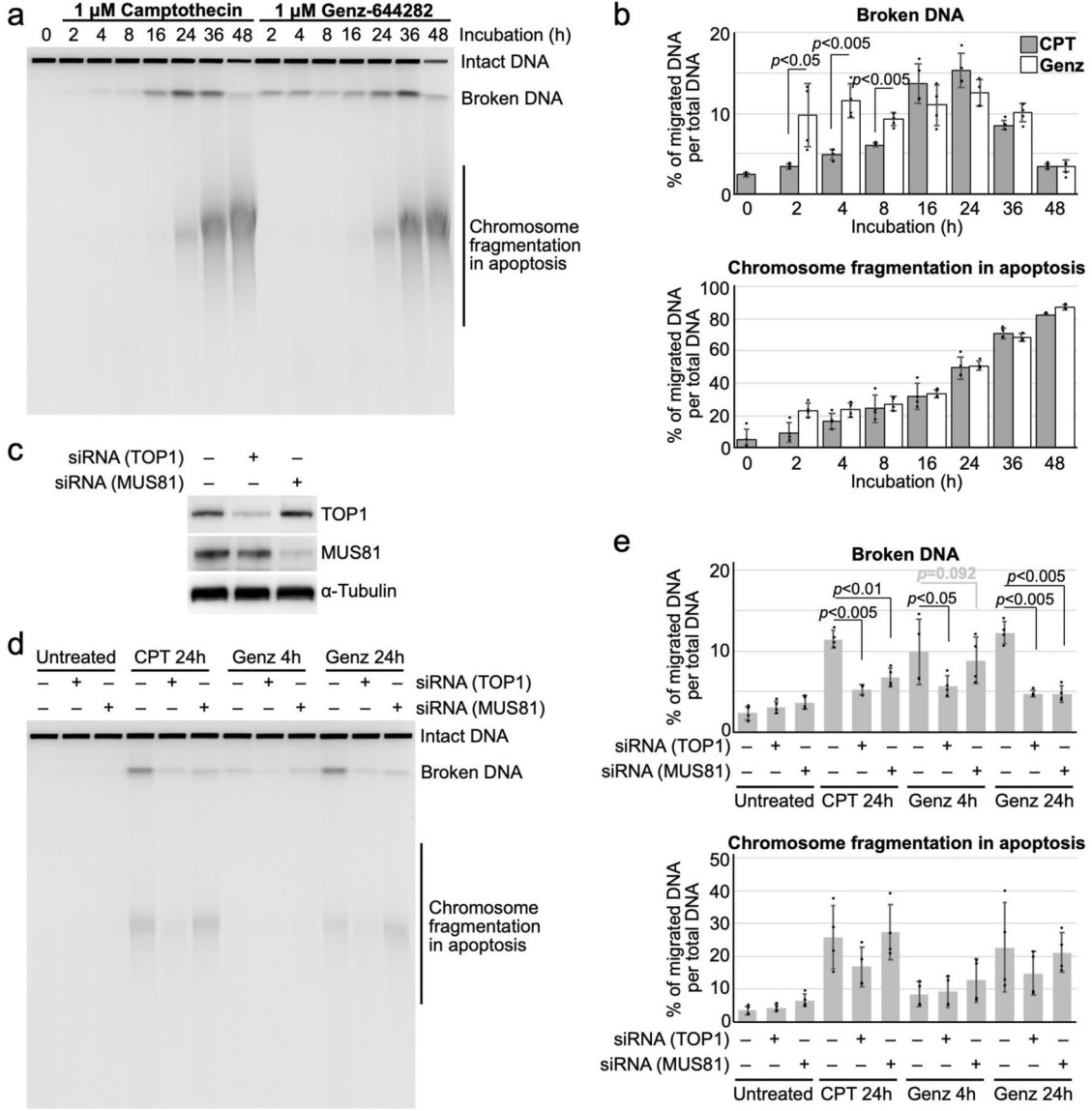

**Fig. 2 Analysis of DSB formation by PFGE. a** Time course analysis of DSB accumulation after treatment with CPT and Genz. Here, 1 μM CPT and 1 μM Genz were applied for 24 h, and the accumulation of broken DNA and chromosome fragmentation were detected by PFGE. **b** Quantification of broken DNA per total DNA and chromosome fragmentation per total DNA after treatment with CPT and Genz. The means and standard deviations were determined from four independent experiments. **c** Western blot analysis of TOP1 and MUS81 after transfection with siRNA against the *MUS81* gene. **d** PFGE analysis of DSB accumulation after treatment with transient TOP1 and MUS81 depletion by siRNA combined with CPT and Genz. Regarding treatment with CPT, cells were treated with 1 μM CPT for 24 h, whereas for Genz-treatment, cells were treated with 1 μM Genz for 4 and 24 h. **e** Quantification of broken DNA per total DNA after treatment with transient TOP1 and MUS81 depletion by siRNA combined with CPT and Genz. The means and standard deviations were determined from four independent experiments.

aphidicolin could suppress the cytotoxicity of Genz, and found that this was indeed the case upon combined treatment with aphidicolin at a concentration of at least 10 nM (Fig. 1g). These results strongly indicate that cells treated with Genz accumulate DSBs, which triggers apoptosis.

Next, we performed time-course experiments. After treatment with CPT and Genz, the accumulation of broken DNA and chromosome fragmentation in apoptosis were assessed by PFGE. After treatment with CPT, broken DNA was detected from 16 to 48 h. The highest accumulation of broken DNA was observed at 24 h. Chromosome fragmentation in apoptosis gradually increased until 48 h (Fig. 2a, b). In contrast, two distinct peaks of broken DNA were observed after treatment with Genz. The first peak appeared around 4 h and the second one appeared between 24 and 36 h (Fig. 2a, b). This result indicates that there are two distinct

mechanisms of Genz-induced DSB formation. Similar to CPT, chromosome fragmentation in apoptosis gradually increased until 48 h after treatment with Genz (Fig. 2a, b). This result also supports the idea that cells treated with Genz accumulate DSBs, which triggers apoptosis.

To explore whether the Genz-induced DSBs were dependent on the function of TOP1, TOP1 was transiently depleted by siRNA transfection, and the accumulation of DSBs after treatment with Genz and CPT was analysed by PFGE (Fig. 2c-e). Because two distinct mechanisms of Genz-induced DSB formation were observed (Fig. 2a, b), we investigated the accumulations of DSBs after treatment with Genz for 4 and 24 h. The depletion of TOP1 protein through the transfection of an siRNA against TOP1 was confirmed (Fig. 2c). Measurement of the Genz-induced DSB formation under TOP1-depleted conditions showed the reduced

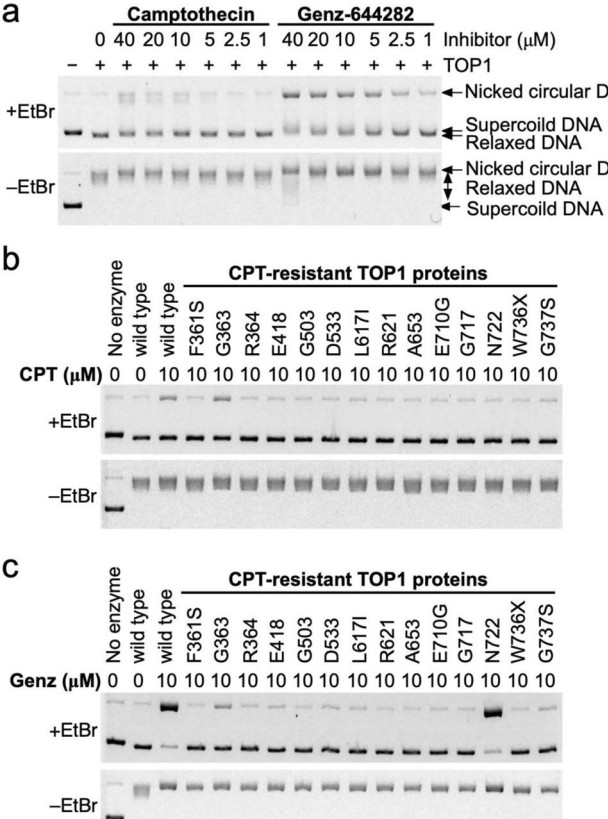

**Fig. 3 Analysis of TOP1 inhibition by CPT and Genz in vitro. a** The detection of DNA–TOP1 complex stabilised during the TOP1 reaction as nicked DNA by agarose gel electrophoresis in the presence of 0.5 μg/mL ethidium bromide (EtBr). The wild-type TOP1 was treated with CPT and Genz. **b** The detection of DNA–TOP1 complex stabilised during the TOP1 reaction as nicked DNA by agarose gel electrophoresis. The wild-type and CPT-resistant TOP1 proteins were treated with 10 μM CPT in the presence of 5 mM MgCl2. **c** The detection of nicked DNA by agarose gel electrophoresis. The wild-type and CPT-resistant TOP1 proteins were treated with 10 μM Genz in the presence of Mg$^{2+}$.

accumulation of DSBs after treatment with Genz for 4 and 24 h (Fig. 2d, e). This indicated that Genz-induced DSB formation was dependent on TOP1 function. In addition, the formation of a majority of CPT-induced DSBs occurred as a result of the cleavage of collapsed DNA replication forks by MUS81–EME1/2 structure-specific endonucleases[4,6]. To address whether Genz-induced DSB formation requires MUS81–EME1/2 function, MUS81 was transiently depleted by siRNA transfection, and the Genz-induced formation of DSBs was analysed by PFGE (Fig. 2d, e). MUS81 depletion reduced the accumulation of Genz-induced DSBs as well as CPT-induced DSBs at 24 h incubation (Fig. 2d–e). However, the accumulation of DSBs induced by treatment with Genz for 4 h was not reduced by MUS81 depletion (Fig. 2d, e). This indicated that Genz-induced DSB formation in the early response occurs without the action of MUS81–EME1/2.

On the basis of these results, we suggest that Genz treatment initially induces TOP1-dependent but MUS81–EME1/2-independent DSB formation in the early response, and a majority of stalled forks are cleaved by MUS81–EME1/2 structure-specific endonuclease in the late response, thus inducing DSBs around DNA replication sites.

**Inhibitory effect of Genz-644282 against CPT-resistant TOP1.** To discover the advantages of Genz over CPT, we next explored

whether Genz could act against CPT-resistant TOP1. Fourteen CPT-resistant mutants—F361S, G363C, R364H, E418K, G503S, D533G, L617I, R621H, A653P, E710G, G717V, N722S, W736X, and G737S—were constructed, and their recombinant TOP1 proteins were purified (Supplementary Fig. 2a, b). First, enzymatic activities of the purified proteins were confirmed by their ability to relax supercoiled DNA (Supplementary Fig. 2c). The activities of G363C, R364H, E418K, D533G, L617I, R621H, E710G, G717V, N722S, and G737S mutants were almost comparable to that of wild-type TOP1, while those of F361S, G503S, A653P, and W736X mutants were reduced by approximately half. Next, we evaluated the CPT resistance of these mutant TOP1 proteins. As mentioned above, CPT exerts its effect by blocking the DNA rejoining step, which results in the accumulation of DNA–TOP1 complexes[24]. These complexes stabilised during the TOP1 reaction were detected as 'nicked DNA' by agarose gel electrophoresis in the presence of 0.5 μg/mL ethidium bromide (EtBr). The accumulation of nicked DNA was not observed for 12 mutants—F361S, R364H, E418K, G503S, D533G, L617I, R621H, A653P, E710G, G717V, N722S, and W736X—in the CPT concentration range from 1 to 100 μM, suggesting that these mutant proteins were resistant to CPT. Meanwhile, the accumulation of nicked DNA after treatment with CPT was observed for two mutants: G363C and G737S (Supplementary Fig. 3). The 50% inhibitory concentrations (IC$_{50}$) of G363C and G737S mutant proteins were 10 and 20 μM, respectively. In contrast, the IC$_{50}$ of wild-type TOP1 was 5 μM (Supplementary Fig. 3). This suggests that G363C and G737S mutations had only slight effects on CPT resistance. On the basis of these results, we set the concentration of CPT and Genz for further analysis as 10 μM because wild-type TOP1 is mostly inhibited at this concentration but mutated TOP1 shows resistance to this level of CPT. The wild-type and mutated TOP1 proteins were treated with CPT and Genz, and the relaxation of supercoiled DNA and accumulation of nicked DNA were examined. The enhanced accumulation of nicked DNA appeared only in the N722S mutant after treatment with Genz (Fig. 3). Meanwhile, slight accumulation of nicked DNA was also observed for the wild type, and the G363C, R364H, E418K, D533G, L617V, E710G, and G737S mutants. No accumulation was observed for the F361S, G503S, A653P, G717V, and W376X mutants.

To assess whether enhanced accumulation of nicked DNA on N722S-mutated protein in vitro correlates with the cytotoxicity of Genz, survival curves after treatment with Genz were determined by the MTT assay using two sets of cell lines: a lymphoblast cell line carrying N722S mutation (CEM/C2) and its control cell line (CCRF-CEM)[19], along with another lymphoblast cell line carrying D533G mutation (CPT-K5) and its control (RPMI8402). The 50% lethality dose (LD$_{50}$) of CEM/C2 to Genz was approximately 5 nM, whereas that of its control, CCRF-CEM, was approximately 1.5 nM (Fig. 4a), with a difference of only 3.3-fold. In contrast, the LD$_{50}$ of CEM/C2 to CPT was around 5 μM and that of CCRF-CEM was around 1.3 nM (Fig. 4b), constituting a difference of approximately 3800-fold. On the basis of this result, we judged that the sensitivity of CEM/C2 to Genz was nearly comparable to that of its control, CCRF-CEM. This suggested that Genz showed cytotoxic effects against the CPT-resistant cell line carrying N722S mutation, which is consistent with a previous study[29]. To investigate whether Genz-induced DSB formation is correlated to its cytotoxicity, the accumulation of DSBs after treatment with CPT and Genz was analysed by PFGE using the CPT-resistant but Genz-sensitve cell line, CEM/C2. CPT-induced DSB formation was not observed but Genz-induced DSB formation was detected in CEM/C2 cells (Fig. 4c, d). Meanwhile, the formation of both CPT- and Genz-induced DSBs was observed in CCRF-CEM cells (Fig. 4c, d). Conversely, the

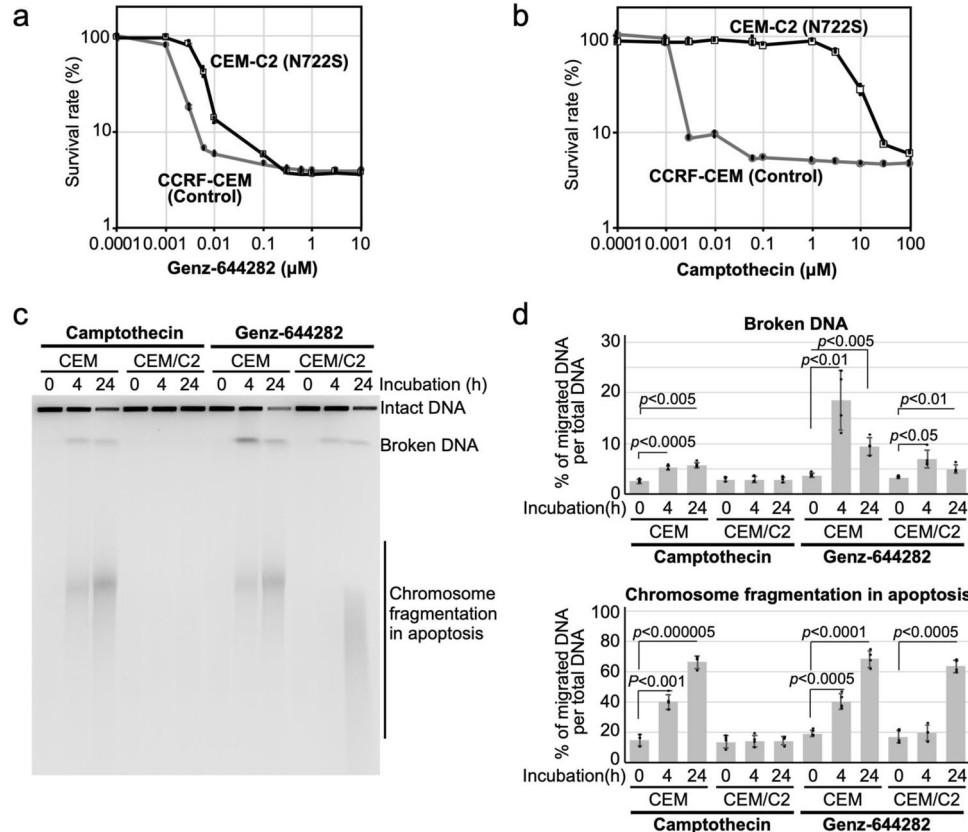

**Fig. 4 Characterisation of the cytotoxic effects of Genz against CPT-resistant cells, CEM/C2 cells, carrying N722S mutations in the TOP1 gene.**
**a**, **b** Survival curve of CCRF–CEM and CEM/C2 after treatment with Genz (**a**) and CPT (**b**). **c** Analysis of DSB formation by PFGE after treatment with CPT and Genz using CPT-resistant cell line CEM/C2 and its control (CCCRF-CEM). Cells were treated with 1 μM CPT and 1 μM Genz for 4 and 24 h.
**d** Quantification of broken DNA per total DNA and chromosome fragmentation per total DNA after treatment with CPT and Genz. The means and standard deviations were determined from four independent experiments.

$LD_{50}$ of CPT-K5 against Genz was around 500 nM, whereas that of its control, RPMI8402, was about 1.2 nM (Fig. 5a), with the difference being around 420-fold. Furthermore, the $LD_{50}$ of CEM/C2 to CPT was approximately 200 nM and that of RPMI8402 was over 100 μM (Fig. 5b), with the difference being over 500-fold. This indicated that cells carrying D533G mutation were resistant to Genz, compared with its control cell line. This in turn suggested that Genz did not exert cytotoxicity against the CPT-resistant cell line carrying D533G mutation. Next, the accumulation of DSBs after treatment with CPT and Genz was analysed by PFGE using the CPT- and Genz-resistant cell line, CPT-K5. Because the survival rates of CPT-K5 after treatment with 0.1 μM and 1 μM Genz were 100% and 27%, and those of RPMI-8402 were 3.6% and 3.8%, respectively (Fig. 5a), RPMI-8402 and CPT-K5 cells were treated with 0.1 μM and 1 μM Genz, and the accumulation of DSBs was investigated by PFGE. DSB formation was observed in RPMI-8402 cells both after treatment with both 0.1 μM and 1 μM Genz (Fig. 5b, e). Meanwhile, no DSB accumulation was observed after treatment with 0.1 μM Genz, and slightly but significantly higher accumulation of DSBs was observed after treatment with 1 μM Genz for 4 h in CPT-K5 cells (Fig. 5b, e). In contrast, CPT-induced DSB formation was observed in RPMI-8402 cells but not in CPT-K5 cells (Fig. 5d, f). Because the survival rate of RPMI-8402 after treatment with 1 μM CPT was 24% and that of CPT-K5 was 99%, the accumulation of DSBs after treatments with CPT was correlated to its cytotoxicity (Fig. 5b, d, f). These results suggested that DSB formation induced by the treatment with CPT and Genz must be a major cause of cell death. Moreover, this result also indicated that the

slight accumulation of nicked DNA after treatment with Genz as appeared in the in vitro experiment shown in Fig. 3 is not sufficient to exert a cytotoxic effect at the cellular level.

On the basis of these results, we conclude that Genz can act against only TOP1 carrying N722S mutation at the molecular and cellular levels.

**Molecular docking study.** Previously, the binding modes of nitidine derivatives, which have structural similarity to Genz, on the TOP1-DNA complex were investigated[36]. In the model, the nitidine derivative 19a was directly held by the arginine at position 364 of TOP1, and this stable interaction interfered with the re-ligation step of the nicked DNA by TOP1[36] (Supplementary Fig. 4a). Therefore, we investigated the binding modes between Genz and TOP1 by a molecular docking method. The result indicated that Genz was intercalated in the DNA strand and was directly held by the arginine at position 364 and aspartate at position 533 of TOP1 (Fig. 6a). However, nitidine was intercalated in the opposite direction of the previously reported model[36] (Fig. 6a, Supplementary Fig. 4a). In our binding model, one of the methoxy groups of Genz would interfere the action of active tyrosine at position 723, which might cause a defect of re-ligation of TOP1 (Fig. 6a). In N722S mutants, the angle of intercalation of Genz in the DNA was changed (Fig. 6b). In this case, two methoxy groups could interfere with the action of active tyrosine, which would exert a strong inhibitory effect on the re-ligation step of TOP1. Meanwhile, our in silico analysis could reproduced the previous finding about the CTP binding on TOP1. CPT intercalated in the DNA strand was directly held by

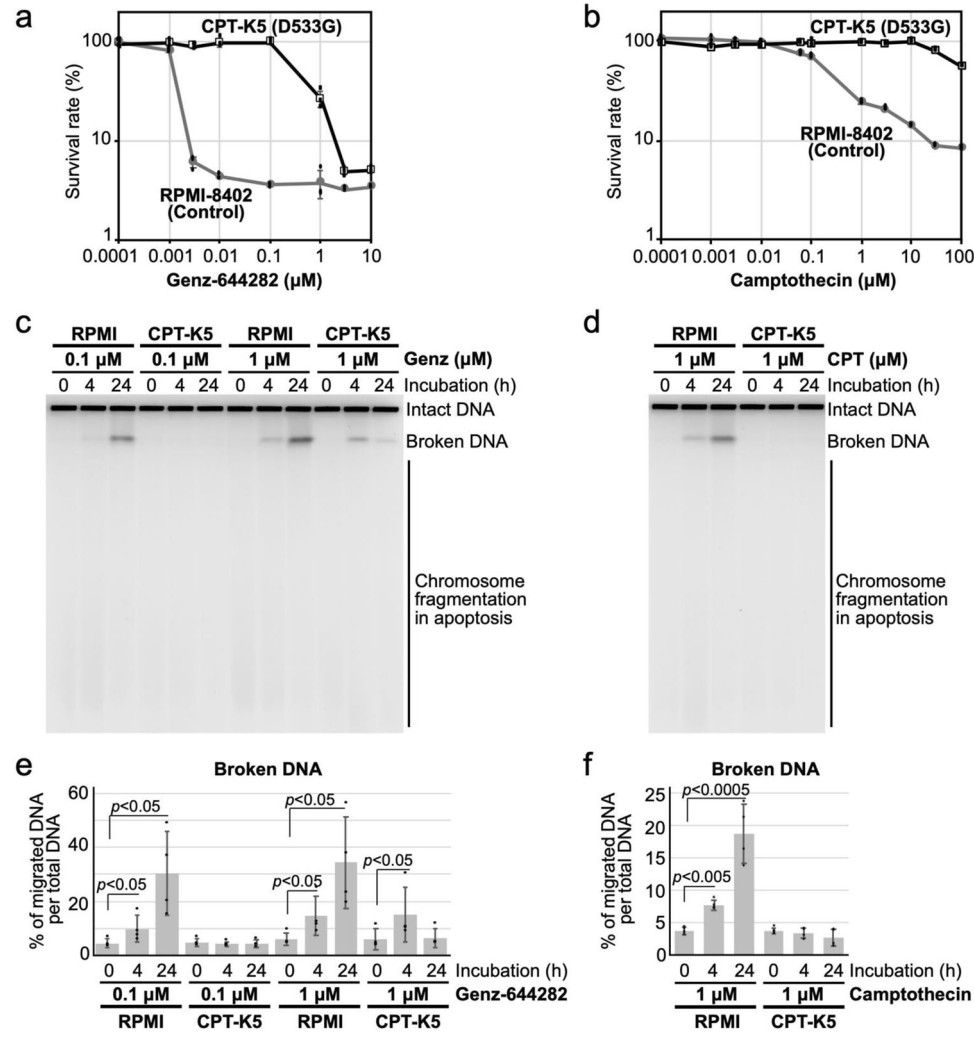

**Fig. 5 Characterisation of the cytotoxic effects of Genz against CPT-resistant cells, CPT-K5 cells, carrying D533G mutation in the TOP1 gene.**
**a**, **b** Survival curve of RPMI-8402 and CPT-K5 after treatment with Genz (**a**) and CPT (**b**). **c** Analysis of DSB formation by PFGE after treatment with Genz. Cells were treated with 0.1 μM and 1 μM Genz for 4 and 24 h. **d** Analysis of DSB formation by PFGE after treatment with CPT. Cells were treated with 1 μM CPT for 4 and 24 h. **e**, **f** Quantification of broken DNA per total DNA and chromosome fragmentation per total DNA after treatment with Genz (**e**) and CPT (**f**). The means and standard deviations were determined from four independent experiments.

the arginine at position 364 and aspartate at position 533 of TOP1, and showed the water-bridged interaction with the asparagine at position 722 (Fig. 6c). N722S mutations abolished the water-bridged interaction CPT (Fig. 6d). Therefore, CPT can no longer interfere with the re-ligation step of TOP1. This model is consistent with the previous model[24,37,38].

**Role of 364th residue of TOP 1 on the interaction with Genz-644282.** Because the simulation study provided a possibility that the arginine residue 364 of TOP1 appears to be crucial for the direct interaction with Genz, and therefore we generated TOP1 with R364H N722S double mutations and analysed the associated activity. The recombinant protein was purified (Supplementary Fig. 5a) and its activity was confirmed by relaxation assay (Supplementary Figs. 5b and 6a, b). Unexpectedly, the catalytic activity of TOP1 with R364H N722S double mutations was approximately half of that of the wild type and of R364H, and N722S mutants (Supplementary Figs. 3c and 5b). Using this finding, wild-type TOP1, R364H, N722S, and double-mutant proteins were incubated with negatively supercoiled DNA in the presence of CPT and Genz and their relaxation activities were investigated. Slight accumulation of nicked DNA was observed

for the wild-type TOP1, but not among the mutated proteins, after treatment with CPT (Fig. 7a, b). Enhanced accumulation of nicked DNA observed for N722S was abolished by the additional mutation of R364H, and the accumulation of nicked DNA was comparable to that for the R364H single mutation, which was less than that of the wild type (Fig. 7c, d). These results confirm that the interaction with arginine residue 364 plays a crucial role in the activity of Genz.

**Role of 722nd residue of TOP1 on the drug resistance of TOP1 to CPT and Genz-644282.** Previous studies suggested that most CPT-resistant mutations, including N722S, impair the interaction between TOP1 and CPT[26]. Here, we showed that treatment with Genz enhanced the accumulation of nicked DNA in TOP1 N722S protein (Fig. 3). Because asparagine residue 722 is located close to an active tyrosine, the role of this residue in accessibility of Genz to the active tyrosine may differ from that of CPT. To understand this, residue 722 was replaced by a small amino acid with a neutral side chain, alanine (N722A); an amino acid with a negatively charged side chain, aspartic acid (N722D); and an amino acid with a positively charged side chain, lysine (N722K). The associated inhibitory effects of CPT and Genz on mutated

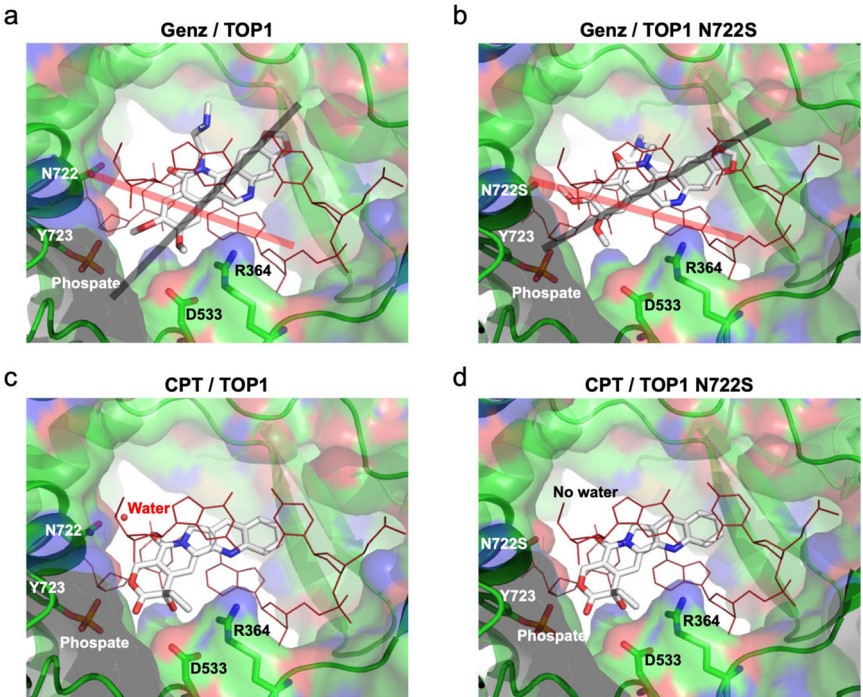

**Fig. 6 Molecular Docking study.** Possible binding models of Genz (**a**) and CPT (**c**) docked in the topotecan binding site of TOP1 and nicked DNA complex using AutoDock Vina. Possible binding model of Genz (**b**) and CPT (**d**) docked in the topotecan binding site of TOP1 N722S and nicked DNA complex TOP1 are shown as ribbons and surfaces. CPT and Genz are shown as sticks. DNA is shown in red lines. Angles between DNA and Genz are indicated by grey and pick lines.

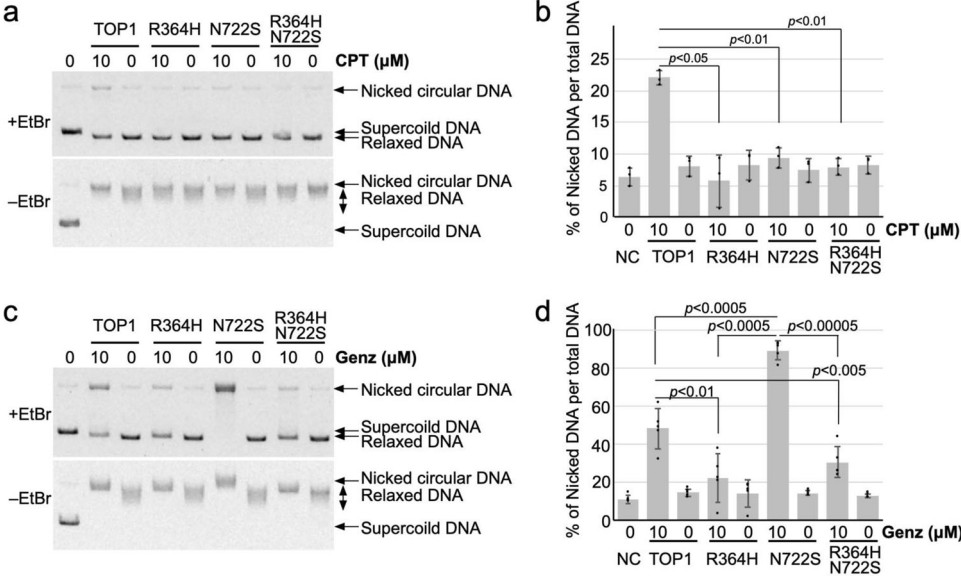

**Fig. 7 Analysis of TOP1 with R364H N722S double mutations. a–d** Biochemical analysis of TOP1 with R364H N722S double mutations. The DNA–TOP1 complex stabilised during the TOP1 reaction was detected as nicked DNA by agarose gel electrophoresis in the presence of $0.5\ \mu g/mL$ EtBr. Relaxation of supercoiled DNA by TOP1 was detected by agarose gel electrophoresis in the absence of EtBr. **a** The wild-type TOP1, R364H, N722S, and R364H N722S double-mutated proteins were treated with CPT, and nicked DNA was detected by agarose gel electrophoresis in the presence of EtBr. **b** Quantification of nicked DNA per total DNA after treatment with CPT. The means and standard deviations were determined from three independent experiments. **c** The wild-type TOP1, R364H, N722S, and R364H N722S double-mutated proteins were treated with Genz, and nicked DNA was detected by agarose gel electrophoresis in the presence of EtBr. **d** Quantification of nicked DNA per total DNA after treatment with Genz. The means and standard deviations were determined from five independent experiments.

TOP1 were compared. Recombinant proteins were purified together with the wild-type and N722S proteins (Supplementary Fig. 6a), and their activities were confirmed by relaxation assay (Supplementary Fig. 6b). The catalytic activity of N722A protein was quite similar to those of the wild-type and N722S proteins (Supplementary Fig. 6b–d). However, the catalytic activities of N722D and N722K proteins were reduced to one-three hundredth of that of the wild type (Supplementary Fig. 6a, e, f). These

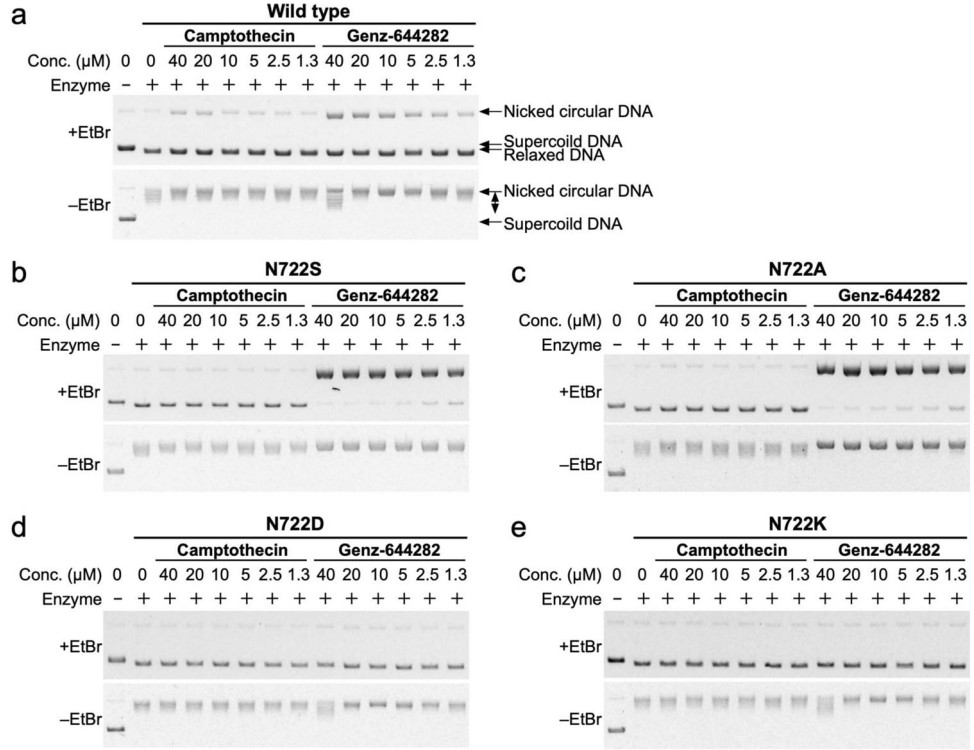

**Fig. 8 Analysis of TOP1 inhibition by CPT and Genz in vitro.** The DNA–TOP1 complex stabilised during the TOP1 reaction was detected as nicked DNA by agarose gel electrophoresis in the presence of 0.5 μg/mL EtBr. Relaxation of supercoiled DNA by TOP1 was detected by agarose gel electrophoresis in the absence of EtBr. **a** The wild-type TOP1 was treated with CPT and Genz, and nicked DNA was detected by agarose gel electrophoresis in the presence of EtBr. (**b–e**) TOP1 N722S (**b**), N722A (**c**), N722D (**d**), and N722D proteins (**e**) were treated with CPT and Genz, and nicked DNA was detected by agarose gel electrophoresis in the presence of EtBr.

results suggest that amino acids with positively and negatively charged side chains are not preferable as the residue 722 of TOP1 with regard to TOP1's catalytic activity. Next, the wild-type and mutated proteins were incubated with negatively supercoiled DNA in the presence of CPT and Genz and their relaxation activities were analysed. For TOP1 with N722A and N722S, supercoiled DNA was completely relaxed without the accumulation of nicked DNA after treatment with CPT, whereas almost all DNA was nicked after treatment with Genz (Fig. 8a–c). The accumulation of nicked DNA on N722A and N722S proteins after treatment with Genz was higher than that on the wild type (Fig. 8a–c). This suggests that the inhibitory effect of Genz on the re-ligation step of TOP1 carrying N722A or N722S mutation is even stronger than that of wild-type TOP1. Next, N722D and N722K proteins were treated with either CPT or Genz. Supercoiled DNA was completely relaxed and no nicked DNA accumulated (Fig. 8a, d, e). This suggests that TOP1 carrying N722D or N722K mutation is resistant to both CPT and Genz.

Overall, we conclude that treatment with Genz exerts cytotoxicity on proliferating cells by introducing DSBs around DNA replication sites, and the advantage of Genz is that it acts against CPT-resistant TOP1 via mutation at the 722nd residue, such as N722A and N722S.

## Discussion

Treatment with CPT causes DNA replication arrest due to a defect in the relaxation of topological stress in front of the DNA replication fork[4]. To remove this stress, MUS81–EME1/2 performs cleavage at stalled DNA replication forks and introduces DSBs at DNA replication sites[4]. Because MUS81–EME1/2 is closely linked to DSB repair via homologous recombination, MUS81–EME1/2-dependent DSBs are promptly repaired[39–43]. However, in the presence of CPT,

repair of DSBs by homologous recombination is also impaired due to topological stress, which results in the accumulation of DSBs, causing cell death. A previous study suggested that treatment with Genz also increased the appearance of γ-H2AX foci, but the mechanism by which Genz treatment induced DSBs was not investigated[29]. In this study, we initially explored the mechanism of Genz-induced DSB formation. Similar to CPT, Genz-induced DSB formation was also suppressed by co-treatment with aphidicolin (Fig. 1c–e), and the cytotoxic effect of Genz was suppressed by co-treatment with aphidicolin, which is also similar to the case for CPT (Fig. 1f, g). In this study, we found two different mechanisms of Genz-induced DSB formation. One was the early response, which occurred within 8 h, requiring TOP1 function but not MUS81–EME1/2 function, while the other was the late response, which appeared later than at 16 h of incubation, requiring both TOP1 and MUS81–EME1/2 functions (Fig. 2c, d). In this context, a question arises about the mechanism through which DSB formation occurs after treatment with Genz. Because Genz could stabilise the TOP1–DNA complex (Figs. 3, 7, 8), one possibility is that a single-strand break, formed as a result of stabilisation of the TOP1-DNA complex by treatment with Genz, might be converted into a DSB through progression of the DNA replication fork. The majority of Genz-induced DSB formation in the early response likely occurs through this mechanism (Fig. 9a). Meanwhile, Genz-induced DSB formation in the late response required the action of MUS81–EME1/2 function (Fig. 2c, d). In this case, stalled replication forks were initially induced by the inhibition of TOP1 function by treatment with Genz, and then some of the stalled replication forks were cleaved by a structure-specific endonuclease, MUS81–EME1/2, which results in the formation of DSBs (Fig. 9a).

Next, we explored the advantages of Genz over CPT. To date, various CPT-resistant mutations in the TOP1 gene have been

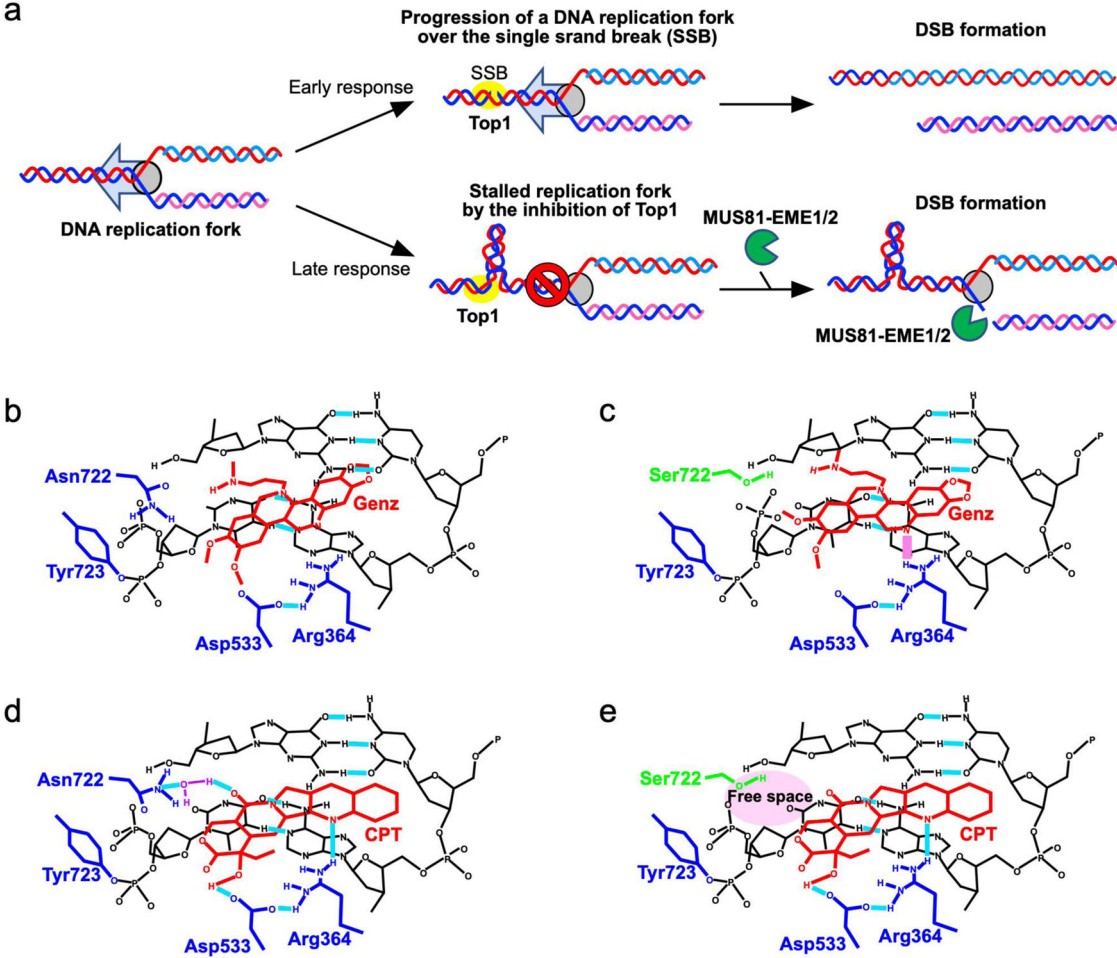

**Fig. 9 Possible model. a** Possible model of CPT-and Genz-induced DSB formation at the DNA replication fork. **b** Action of Genz on the wild-type TOP1. Genz was intercalated in DNA strand and was directly held by the arginine at position 364 and aspartate at position 533 of TOP1. One of the methoxy groups of Genz would interfere with the re-ligation step of TOP1. **c** Action of Genz on CPT-resistant TOP1 carrying N722S mutation. In N722S mutant, the angle of intercalation of Genz in the DNA was changed. In this case, two methoxy groups could interfere with the action of the active tyrosine residue, which resulting in severe defect of the re-ligation of TOP1. **d** Action of CPT on the wild-type TOP1. The water-bridged interaction between asparagine residue 722 and CPT inhibits the re-ligation of cleaved DNA. **e** Mechanism of CPT resistance of TOP1 carrying N722S mutation. Impaired interaction between residue 722 and CPT provides free space for re-ligation of the phosphodiester bond of the DNA. **b–e** The structure of the DNA is represented by black lines and letters. Residues of TOP1 protein are represented by blue and green lines and letters. Inhibitors, CPT and Genz, are represented by red lines and letters. Hydrogen bonds between TOP1 and the inhibitor are represented by light blue lines. Additional factors, such as water and phosphate, are represented by purple lines and letters.

reported. We therefore purified 14 recombinant proteins and examined the accumulation of nicked DNA after treatment with Genz. Among these, only TOP1 carrying the N722S mutation showed the accumulation of nicked DNA, while the rest showed a reduction of it (Fig. 3). This indicated that the advantage of Genz appeared only for TOP1 mutated at the 722nd residue. To understand the role of this residue in drug resistance, N722A, N722D, and N722K mutated proteins were additionally constructed and purified, for which the effects of Genz and CPT were examined. The effects of these drugs on TOP1 carrying the N722A mutation were comparable to those on TOP1 carrying the N722S mutation (Fig. 9a–c). Meanwhile, no nicked DNA accumulated after treatment with Genz and CPT for N722D and N722K mutant proteins (Fig. 9d, e). This suggests that Genz is no longer advantageous for N722D and N722K mutant proteins. However, the catalytic activities of N722D and N722K mutant proteins were 300-fold lower than those of the wild-type, and N722S and N722A mutant proteins (Supplementary Fig. 6). Given that TOP1 is essential for cell growth, whether N722D and

N722K mutations maintain growth advantages for cancer cells is unclear and requires further investigation. One central question is why the enhanced accumulation of nicked DNA after treatment with Genz is observed in TOP1 carrying N722S and N722A mutations. This accumulation was markedly higher than that of the wild type. This implies that Genz has greater access to the active site of TOP1 in N722S and N722A mutant proteins than in the wild type. Previously, the binding modes of CPT and a nitidine derivative that has structural similarity to Genz were investigated[36]. Therefore, we investigated the modes of binding between Genz and TOP1 by a molecular docking method. Genz was intercalated in the DNA strand and was directly held by the arginine at position 364 and aspartate at position 533 of TOP1 (Fig. 6b), but the binding mode was differed from that of nitidine derivative 19a, as previously reported (Supplementary Fig. 4a). In our binding model, one of the methoxy groups of Genz would interfere with the re-ligation step of TOP1 (Figs. 6a and 7c, d). In N722S mutations, amino acid residues with a small side chain, such as alanine and serine, do not interfere with the intercalation

of Genz in the DNA strand by π-π interaction, which allows alteration of the angle at which Genz is intercalated in DNA (Fig. 6b). In this binding mode, both of the two methoxy groups can interfere with the action of the active tyrosine residue, resulting in severe defect of the re-ligation of TOP1. This might cause strong interference of the re-ligation by Genz in N722S and N722A mutants, compared with the case in wild-type TOP1. Our binding model of Genz on TOP1 differed from that previously reported using the nitidine derivative 19a. That binding model is presented in Supplementary Fig. 4a, which can explain how Genz interferes with the re-ligation of TOP1. Because the nitidine derivative interacts with TOP1 through an amino group arginine at position 364 and asparagine at 722, there is a possibility that Genz interacts with TOP1 through an amino group arginine at position 364 and asparagine at 722. These interactions contribute to interference with the re-ligation step (Supplementary Fig. 4a). However, owing to the structural difference, the impaired interaction between Genz and residue 722 caused by N722S or N722A mutation does not provide sufficient space. When an amino acid residue with a small side chain is present at this position, Genz deeply invades the space between the two ends of the nicked DNA that are created as an intermediate structure of the TOP1 reaction because the side chain of residue 722 cannot act as a physical barrier to it (Supplementary Fig. 4b). This causes strong inhibition of the re-ligation step, resulting in the enhanced accumulation of nicked DNA. To understand the precise action of Genz, further biochemical and structural studies would be required. Nonetheless, Genz does not lose its inhibitory effect on CPT-resistant TOP1 carrying N722S or N722A mutation. In contrast, CPT interacts with the asparagine residue at position 722 via water-bridge interaction, which physically interferes with re-ligation from a DNA–TOP1 complex (Figs. 6b, 9d). CPT–TOP1 interaction via residue 722 is abolished by N722S or N722A mutation (Figs. 6d, 9e). This impaired interaction provides sufficient free space to catalyse the re-ligation step.

Overall, we identified two features of Genz. One is that treatment with Genz induced TOP1-dependent DSB formation around DNA replication sites, and MUS81–EME1/2 is only required for DSB formation in the late response. This contributed to cytotoxicity against proliferating cells. The other is that the role of residue 722 of TOP1 differs between CPT and Genz. This difference conferred the advantages of Genz over CPT regarding the cytotoxic activity to CPT-resistant cell lines caused by N722S and N722A mutations.

## Methods

**Cell lines, media, and compounds**. All human cells were cultivated at 37 °C and 5% CO$_2$. SV40-transformed human fibroblasts MRC5sv[32] were cultivated in Dulbecco's Modified Eagle's Medium (DMEM) with 10% (v/v) foetal bovine serum (FBS). Human lymphoblast cell lines, namely, CCRF-CEM (ATCC CCL-119)[6], CEM/C2 (ATCC CRL-2264)[6], RPMI8402[6], and CPT-K5[6,44], were cultivated in RPMI1640 medium with 10% FBS. Insect cells, High Five, were cultivated in Grace's insect medium with 10% (v/v) FBS at 28 °C. Camptothecin (CAS No. 7689-03-4, PubChem CID: 2538) was purchased from FujiFilm Co Ltd., and Genz-644282 (CAS No. 529488-28-6, PubChem CID: 10294813) was purchased from Selleck Chemicals Co Ltd and MedChemExpress (MCE) Co Ltd.

**Immunofluorescent staining**. Immunofluorescent staining was performed as previously described[6]. Subconfluent MRC5sv cells on coverslips were treated with the indicated concentrations of CPT and Genz for 1, 2, 4, 8, 16, and 24 h. Thereafter, cells were fixed with 2% (w/v) paraformaldehyde in phosphate-buffered saline (PBS) for 15 min. Cells were then permeabilised using 0.25% (v/v) Triton X-100 in PBS. Rabbit polyclonal anti-53BP1 antibody (1:300; Novus Biologicals) and mouse monoclonal anti-γ-H2AX antibody (1:500; Millipore) were used as the primary antibodies. CF488-conjugated donkey anti-mouse antibody (1:400; Biotium) and AF555-conjugated donkey anti-mouse antibody (1:400; Biotium) were used as the secondary antibodies.

Ethynyldeoxyuridine (EdU) staining was performed as previously described[32]. Subconfluent cells on coverslips were treated with 10 μM EdU for 15 min. Then, the medium was replaced with DMEM containing 10% FBS. Subsequently, cells

were incubated with 0.1 μM or 1 μM CPT and Genz for either 2 or 4 h. After treatment, cells were fixed with 2% paraformaldehyde in PBS for 15 min. Cells were then permeabilised with 0.25% (v/v) Triton X-100 in PBS. EdU was visualised using a ClickIT EdU imaging kit (Life Technologies). Briefly, cells on the coverslips were washed with 0.5% BSA in PBS and treated with 0.25% (v/v) Triton X-100 in PBS for 20 min. The conjugation of Alexa-488 on EdU was performed in the reaction buffer provided with this kit for 30 min at room temperature. After the reaction, cells were washed with 0.5% (w/v) BSA in PBS. Rabbit polyclonal anti-53BP1 antibody (1:300; Novus Biologicals) and mouse monoclonal anti-γ-H2AX antibody (1:500; Millipore) were used as the primary antibodies. CF555-conjugated donkey anti-mouse antibody (1:400; Biotium) and CF633-conjugated donkey anti-mouse antibody (1:400; Biotium) were used as secondary antibodies.

Nuclear staining patterns were visualised using Vectashield mounting medium with DAPI (Vector Laboratories). Cells were visualised using a fluorescent microscope (Leica TCS SP8).

**Pulse field gel electrophoresis (PFGE)**. The details of the method of PFGE applied here were as previously described[45,46]. Subconfluent cultures of MRC5sv were treated with the indicated compounds for 24 h. Thereafter, cells were harvested using trypsinisation, and plugs [0.5% (w/v) agarose containing $2.5 \times 10^5$ cells in PBS] were prepared with a CHEF disposable plug mould (Bio-Rad). The plugs were incubated in lysis buffer [100 mM EDTA, 1% (w/v) lauryl sarcosine sodium, 0.2% (w/v) sodium deoxycholate, 1 mg/mL proteinase K] at 37°C for 24 h, and then washed with TE buffer [10 mM Tris-HCl (pH8.0), 100 mM EDTA]. PFGE was performed at 13 °C for 23 h in 0.9% (w/v) agarose containing 0.25×TBE buffer using a Biometra Rotaphor (Analytik Jena). The parameters were as follows: voltage, from 180 to 120 V log; angle, from 120° to 110° linear; and intervals, from 30 to 5 s log. The gel was stained with 0.5 μg/ml ethidium bromide (EtBr) in 0.25×TBE buffer and analysed using a Typhoon FLA7000 scanner (GE Healthcare). Semi-quantitative analysis was performed using ImageQuant (GE Healthcare). The percentages of 'Broken DNA' were calculated from the amount of broken DNA relative to total DNA (intact DNA + broken DNA + chromosome fragmentation in apoptosis), and those of 'Chromosome fragmentation in apoptosis' was calculated from the amount of chromosome fragmentation in apoptosis relative to total DNA.

**Colony survival assay**. Five hundred MRC5sv cells were plated in 60 mm dishes. After 6 h, the indicated compounds were added and cells were incubated for 24 h. Subsequently, cells were washed with PBS and incubated with fresh DMEM with 10% (v/v) FBS for 7 days. Cells were again washed with PBS and stained with staining buffer [0.1% (w/v) Coomassie brilliant blue R-250, 50% (v/v) methanol, 7% (v/v) acetate]. Subsequently, the number of colonies was counted. The survival rate was calculated by determining the number of colonies that appeared after the treatment relative to that without treatment. Means and standard errors were determined using triplicate data.

**siRNA and western blotting**. Commercial human Topo I siRNA was used in this study (human Top1 siRNA, sc-36694; Santa Cruz Biotechnology). TOP1 was analysed by western blotting using a mouse anti-Topo I antibody (C-21) (1:10,000, sc-32736; Santa Cruz Biotechnology). Commercial human MUS81 siRNA was used in this study (human MUS81 siRNA, L-016143-01-0005; Horizon). MUS81 was analysed by western blotting using a mouse anti-MUS81 antibody (MTA30 2G10/3) (1:10,000, ab-14387; Abcam). As a loading control, α-tubulin was analysed with mouse anti-α-tubulin antibody (10G10) (1:10,000, 017-25031; Fujifilm). Horse-radish peroxidase (HRP)-conjugated donkey anti-mouse IgG (1:10,000, 715-035-150; Jackson ImmunoResearch Laboratories) and HRP-conjugated anti-rabbit IgG (1:10,000, 711-035-152; Jackson ImmunoResearch Laboratories) were used as secondary antibodies. The bands were visualised with Chemi-Lumi One Super (Nacalai Tesque) and analysed with LAS4000 Mini (GE Healthcare Life Science).

**MTT assay**. The proliferative activity of the human lymphoblast cell lines RPMI8402, CPT-K5, CCRF-CEM, and CEM/C2 was determined in triplicate using Cell Proliferation Kit I (Roche), following the manufacturer's instructions. Briefly, two pairs of cell lines—CEM-C2/CCRF-CEM and RPMI8402/CPT-K5—were seeded into 96-well plates at a density of $1 \times 10^4$ cells/well and incubated at 37 °C with 5% CO$_2$ for 1 h. Subsequently, 50 μL of CPT and Genz-644282 was added to each well. After 72 h of incubation, 10 μL of MTT [3-(4,5-dimethylthiazol-2-yl)-2,5-diphenyltetrazolium bromide] labelling reagent was added to each well and the plate was incubated for 4 h. Thereafter, 100 μL of solubilisation solution was added and the plates were maintained overnight in an incubator. A cell proliferation curve was then created from the results of the baseline-corrected absorbance measurements at 570 and 650 nm.

**Purification of recombinant TOP1**. His-tagged TOP1 was purified using a baculovirus-insect cell expression system. The activity of the 6×His-tagged protein was previously confirmed[6]. The baculovirus, which carried wild-type and mutated TOP1, was infected into High Five cells and then the cells were incubated for 48 h. After induction, the cells were collected in a 50-mL tube and lysed using lysis buffer [50 mM sodium phosphate buffer (pH 7.0), 1 M NaCl, 1 mM β-mercaptoethanol,

0.1% (v/v) Nonidet P-40, a complete tablet containing a cocktail of inhibitor proteinases (Roche)] on ice for 1 h. The lysate was clarified by centrifugation at 60,000 rpm (Beckman; 70.1 Ti) and 4 °C for 1 h, and the supernatant was subsequently incubated with TALON resin (GE Healthcare) for 30 min. After incubation, the resin was transferred to a disposable column (Bio-Rad) and washed with 10 bed volumes of HS wash buffer [50 mM sodium phosphate buffer (pH 7.0), 800 mM NaCl, 1 mM β-mercaptoethanol] and IM wash buffer [50 mM sodium phosphate buffer (pH 7.0), 500 mM NaCl, 1 mM β-mercaptoethanol, 10 mM imidazole]. The bound proteins were eluted with elution buffer [50 mM sodium phosphate buffer (pH 7.0), 300 mM imidazole, 300 mM NaCl, 1 mM β-mercaptoethanol]. The elution fraction was then loaded on a HiTrap Heparin HP column (GE Healthcare) with buffer A [25 mM HEPES-Na (pH 7.5), 150 mM NaCl, 5 mM β-mercaptoethanol, 10% (v/v) glycerol]. The bound protein was eluted in buffer B [25 mM HEPES-Na (pH 7.5), 1 M NaCl, 5 mM β-mercaptoethanol, 10% (v/v) glycerol] with a linear gradient of 0.15 to 1 M NaCl. The purified protein was confirmed by SDS-PAGE.

**Relaxation assay**. The method for the relaxation assay followed that previously reported[6]. The purified TOP1 was diluted based on its activity with buffer B [25 mM HEPES-Na (pH 7.5), 1 M NaCl, 5 mM β-mercaptoethanol, 10% (v/v) glycerol]. Subsequently, 0.25 μg of supercoiled pBR322 DNA in 10 μL was relaxed with TOP1 in reaction buffer [35 mM Tris-Cl (pH 8.0), 72 mM KCl, 5 mM MgCl$_2$, 5 mM dithiothreitol (DTT), 5 mM spermidine, 0.01% (w/v) BSA]. The indicated compound was also added and the mixture was incubated at 37 °C for 15 min. To stop the reaction, 1/10 volume of 1 mg/mL proteinase K was added and the mixture was incubated at 37 °C for 5 min. After the reaction, the DNA was analysed by 0.8% agarose gel electrophoresis with TBE buffer in the absence of EtBr. Under these conditions, the upper band represents relaxed circular DNA and/or nicked circular DNA, and the intermediate and lower bands represent supercoiled DNA[6,45]. After electrophoresis, the gel was stained with EtBr and analysed using a Typhoon FLA7000 scanner (GE Healthcare). Nicked DNA was analysed by 0.8% (w/v) agarose gel electrophoresis with TBE buffer in the presence of 0.5 μg/mL EtBr. Under these conditions, the upper band represents nicked circular DNA and the lower band represents covalently closed circular DNA[6].

**Molecular docking study**. Three-dimensional structures of camptothecin (CPT) and Genz-644282 (Genz) were generated and energy-minimized with Avogadro software 1.2.0[47]. CPT and Genz were subjected to a docking study with a human DNA topoisomerase I (TOP1) complex with nicked DNA, which was adopted from the crystal structure (PDB ID: 1K4T) after removing inhibitors and water. For each compound, nine docking models were generated using Autodock Vina software 1.1.2[48] and the highest-affinity (lowest-energy) model is shown in the Fig. 6. The graphics were created with the PyMol Molecular Graphics System (Schrodinger, LLC). The TOP1 N722S structure was generated in the PyMol software by replacing the side chain of Asp722 with that of serin and subjected to docking study.

**Statistics and reproducibility**. Means and SEMs represented survival curves in Figs. 1f, g, 4a, b, 5a, b was performed from results of three independent experiments. Means and SDs of PFGE results represented in Figs. 1d, e, 2b, e, 4d, 5e, f was performed from results of four independent experiments. Means and SDs of biochemical result of CPT-treatment represented in Fig. 7b was performed from results of three independent experiments and that of Genz-treatment represented in Fig. 7d was done from results of five independent experiments. All P-values were evaluated by one-tailed student t-test.

**Reporting Summary**. Further information on research design is available in the Nature Research Reporting Summary linked to this article.

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

## Acknowledgements

We thank Edanz (https://jp.edanz.com/ac) for editing a draft of this manuscript. The MRC5sv cell line was kindly provided by Dr. Roland Kanaar. The RPMI8402 cell line was kindly provided by Dr. Yoshiaki Ohnishi. KH was funded by a Grant-in-Aid for Scientific Research for Young Scientists (A) (25710010) coordinated by the Japan Society for the Promotion of Science (JSPS), The Ministry of Education, Culture, Sports, Science and Technology (MEXT), Japan.

## Author contributions

M.N., T.Tetabayashi, and K.H. designed the study and performed the majority of the experiments. M.N. performed the biochemical and microscopic analyses. T.Tetabayashi, T.O., S.A., T.Tomo, and M.K. performed other cellular analyses. S.M. performed molecular docking study (in silico analysis). K.T., H.T., H.A., T.I., and H.N. supervised the project, and K.H. wrote the manuscript.

## Competing interests

The authors declare no competing interests.
