## [Peer Review File · Communications Biology]

Reviewers' comments:

Reviewer #1 (Remarks to the Author):

In this manuscript, the authors analyze the mechanisms of action of Genz-644282, a non-camptothecin inhibitor of topoisomerase I (TOP1). In particular, they show that the cytotoxic activity of Genz-644282 is mainly related to the induction of replication-dependent DNA double-strand breaks (DSBs), a mechanism of action similar to that of camptothecin. They also show that, unlike camptothecin, Genz-644282 is still able to induce TOP1 trapping and cytotoxicity when TOP1 carries the N722S mutation. The experiments are generally sound and the conclusions are supported by the data. However, the main findings of this manuscript have been previously reported, including that Genz-644282 induces DSBs mostly in S-phase cells and overcomes camptothecin resistance in cells carrying the TOP1 N722S mutation (Sooryakumar et al, Mol Cancer Ther 2011). Therefore, this manuscript is rather incremental/complementary and does not significantly advance our knowledge of the mechanism of action of Genz-644282.

Minor points:

Introduction, line 48: It is now established that camptothecin also induces DSBs outside of S phase by a transcription-dependent mechanism, and that camptothecin is also cytotoxic in non-proliferating cells. This should be mentioned.

Drug concentration and treatment time are sometimes missing from figure legends.

Reviewer #2 (Remarks to the Author):

This manuscript by Nishida et al. focuses on Genz, a non-camptothecin inhibitor of human TOP1, that may overcome disadvantages of camptothecin for cancer therapy. The experiments are to provide information on the mechanism of action of Genz, and to demonstrate the effectiveness of Genz against cell lines with camptothecin (CPT) resistant mutation at residue 722 of TOP1. The experimental results are in general of high quality. Certain properties of camptothecin, including induction of gamma-H2AX followed by 53BP1, replication-dependence for generating DSB and cytotoxicity, could be observed for Genz. It is interesting that the results here showed that while TOP1 with N722S and N722A mutation are resistant to CPT for inhibition of DNA religation, increased TOP1 covalent complex was observed from Genz treatment of the mutant N722S and N722A TOP1. The manuscript also showed that Genz had good cytotoxicity against the CPT-resistant cell line CCRF-CEM with the N722S mutation in TOP1. These results should be of interest for the potential application of Genz in cancer therapy. The results are also relevant for the understanding of human TOP1 interactions with inhibitors that may affect its cleavage-religation of DNA and the accumulation of the TOP1 covalent complex. However, there are significant weaknesses in the manuscript that need to be addressed.

1. According to 2000 review of camptothecin mechanism of action by Leroy Liu and others <https://pubmed.ncbi.nlm.nih.gov/11193884/>: "The primary mechanism of cell killing by CPT is S-phase-specific killing through potentially lethal collisions between advancing replication forks and topo-I cleavable complexes. Collisions with the transcription machinery have also been shown to trigger the formation of long-lived covalent topo-I DNA complexes, which contribute to CPT cytotoxicity." The stabilization of the covalent TOP1-DNA covalent complex (TOP1cc) and associated DNA breaks may be more important than the defect in the relaxation of topological stress in front of the replication fork. It is puzzling that the percent of broken DNA shown from Genz treatment in Figure 2 is much lower than the percent of broken DNA shown from CPT treatment. Yet Genz is 10-fold more potent than CPT for decreasing cell survival (Figure 1 f-g). The treatment of TOP1 siRNA produced a much greater effect on TOP1 DNA breaks from CPT when compared to Genz (Figure 1c). These results suggest that there may be additional mechanism of action for Genz. Such possibility should be considered when stating that "the mechanism of action of Genz at the cellular level is the

same as that of CPT" (line 258).

2. Since it is possible that there is additional target for Genz that is independent of TOP1, selection for Genz-resistance and characterization of the associated mutations are needed to confirm that TOP1 is the major cellular target for Genz. The characterization of Genz resistance would also help establish if Genz resistance arises at similar rate as CPT resistance and if the Genz resistance affects the proliferation rate of the cancer cells. Such information is needed to evaluate Genz as alternative treatment option in the clinic.

3. I cannot find the drug concentrations used to generate the data in Figure 1b-e. These drug concentrations should have been included in the Figure 1 legend.

4. There are a number of instances where use of English needs to be corrected. For example:

i. line 38, "cutting of the phosphate group" – should be linking to the phosphate group

ii. line 78, "found for removing another CPT-resistant cell line"

iii. line 85, "Genz acts to CPT-resistant TOP1"

iv: line 105, "it was suggested that stalled DNA.."

v: line 228, "reduced by one-three hundredth of that", should be "reduced to one-three hundredth of that"

5. The sources of chemicals (Genz and CPT) should be described under Methods.

Reviewer #3 (Remarks to the Author):

In this paper, the authors presented their results about the non-camptothecin topo I inhibitor, Genz-644282. They have compared the effects of camptothecin and Genz in vitro on purified topo I as well as on 14 purified topo I camptothecin-resistant mutants. They have also studied in detail the mechanism of Genz cytotoxicity in cells expressing wild type topo I. Genz toxicity on 14 cell lines each expressing a different camptothecin-resistant mutant has also been evaluated. They found that the mechanism of cytotoxicity of Genz was very similar to that of camptothecin i.e., stalling of replication forks through topo I inhibition, followed by cleavage of the forks by MUS81-EME1/2 structure-specific endonuclease inducing DSBs. This is followed by apoptosis. The most interesting finding (novelty) of the paper stem from the observation that the only camptothecin-resistant topo I mutant, N722S, that is still sensitive to Genz both in vitro and in vivo, generates more nicked DNA than wild-type topo I in the presence of Genz. Based on a previous model in which the stable interaction of nitidine derivatives, having structural similarity to Genz, with arginine 364 and asparagine 722 of topo I inhibits the religation step, the authors have performed mutagenesis studies at these positions. The results allowed them to present a model explaining why the N722S mutation make topo I resistant to camptothecin but not to Genz. Overall the results presented in this paper are clear and they support very well the conclusions. Furthermore, the results of experiments related to asparagine 722 residue may help to design better camptothecin-like topo I inhibitors.

Minor comment:

In the siRNA experiment in Fig. 2, a stronger Broken DNA signal is seen for Genz than for camptothecin in the absence of topo I. Does it mean that Genz is less specific to topo I than camptothecin?

Reviewers' comments:

Reviewer #1 (Remarks to the Author):

In this manuscript, the authors analyze the mechanisms of action of Genz-644282, a non-camptothecin inhibitor of topoisomerase I (TOP1). In particular, they show that the cytotoxic activity of Genz-644282 is mainly related to the induction of replication-dependent DNA double-strand breaks (DSBs), a mechanism of action similar to that of camptothecin. They also show that, unlike camptothecin, Genz-644282 is still able to induce TOP1 trapping and cytotoxicity when TOP1 carries the N722S mutation. The experiments are generally sound and the conclusions are supported by the data. However, the main findings of this manuscript have been previously reported, including that Genz-644282 induces DSBs mostly in S-phase cells and overcomes camptothecin resistance in cells carrying the TOP1 N722S mutation (Sooryakumar et al, Mol Cancer Ther 2011). Therefore, this manuscript is rather incremental/complementary and does not significantly advance our knowledge of the mechanism of action of Genz-644282.

Minor points:

Introduction, line 48: It is now established that camptothecin also induces DSBs outside of S phase by a transcription-dependent mechanism, and that camptothecin is also cytotoxic in non-proliferating cells. This should be mentioned.

Drug concentration and treatment time are sometimes missing from figure legends.

Reviewer #2 (Remarks to the Author):

This manuscript by Nishida et al. focuses on Genz, a non-camptothecin inhibitor of human TOP1, that may overcome disadvantages of camptothecin for cancer therapy. The experiments are to provide information on the mechanism of action of Genz, and to demonstrate the effectiveness of Genz against cell lines with camptothecin (CPT) resistant mutation at residue

722 of TOP1. The experimental results are in general of high quality. Certain properties of camptothecin, including induction of gamma-H2AX followed by 53BP1, replication-dependence for generating DSB and cytotoxicity, could be observed for Genz. It is interesting that the results here showed that while TOP1 with N722S and N722A mutation are resistant to CPT for inhibition of DNA religation, increased TOP1 covalent complex was observed from Genz treatment of the mutant N722S and N722A TOP1. The manuscript also showed that Genz had good cytotoxicity against the CPT-resistant cell line CCRF-CEM with the N722S mutation in TOP1. These results should be of interest for the potential application of Genz in cancer therapy. The results are also relevant for the understanding of human TOP1 interactions with inhibitors that may affect its cleavage-religation of DNA and the accumulation of the TOP1 covalent complex. However, there are significant weaknesses in the manuscript that need to be addressed.

1. According to 2000 review of camptothecin mechanism of action by Leroy Liu and others <https://pubmed.ncbi.nlm.nih.gov/11193884/>: "The primary mechanism of cell killing by CPT is S-phase-specific killing through potentially lethal collisions between advancing replication forks and topo-I cleavable complexes. Collisions with the transcription machinery have also been shown to trigger the formation of long-lived covalent topo-I DNA complexes, which contribute to CPT cytotoxicity." The stabilization of the covalent TOP1-DNA covalent complex (TOP1cc) and associated DNA breaks may be more important than the defect in the relaxation of topological stress in front of the replication fork. It is puzzling that the percent of broken DNA shown from Genz treatment in Figure 2 is much lower than the percent of broken DNA shown from CPT treatment. Yet Genz is 10-fold more potent than CPT for decreasing cell survival (Figure 1 f-g). The treatment of TOP1 siRNA produced a much greater effect on TOP1 DNA breaks from CPT when compared to Genz (Figure 1c). These results suggest that there may be additional mechanism of action for Genz. Such possibility should be considered when stating that "the mechanism of action of Genz at the cellular level is the same as that of CPT" (line 258).

2. Since it is possible that there is additional target for Genz that is independent of TOP1, selection for Genz-resistance and characterization of the associated mutations are needed to confirm that TOP1 is the major cellular target for Genz. The characterization of Genz resistance would also help establish if Genz resistance arises at similar rate as CPT resistance and if the Genz resistance affects the proliferation rate of the cancer cells. Such information is needed to

evaluate Genz as alternative treatment option in the clinic.

3. I cannot find the drug concentrations used to generate the data in Figure 1b-e. These drug concentrations should have been included in the Figure 1 legend.

4. There are a number of instances where use of English needs to be corrected. For example:

i. line 38, "cutting of the phosphate group" – should be linking to the phosphate group

ii. line 78, "found for removing another CPT-resistant cell line"

iii. line 85, "Genz acts to CPT-resistant TOP1"

iv: line 105, "it was suggested that stalled DNA.."

v: line 228, "reduced by one-three hundredth of that", should be "reduced to one-three hundredth of that"

5. The sources of chemicals (Genz and CPT) should be described under Methods.

Reviewer #3 (Remarks to the Author):

In this paper, the authors presented their results about the non-camptothecin topo I inhibitor, Genz-644282. They have compared the effects of camptothecin and Genz in vitro on purified topo I as well as on 14 purified topo I camptothecin-resistant mutants. They have also studied in detail the mechanism of Genz cytotoxicity in cells expressing wild type topo I. Genz toxicity on 14 cell lines each expressing a different camptothecin-resistant mutant has also been evaluated. They found that the mechanism of cytotoxicity of Genz was very similar to that of camptothecin i.e., stalling of replication forks through topo I inhibition, followed by cleavage of the forks by MUS81-EME1/2 structure-specific endonuclease inducing DSBs. This is followed by apoptosis. The most interesting finding (novelty) of the paper stem from the observation that the only camptothecin-resistant topo I mutant, N722S, that is still sensitive to Genz both in vitro and in vivo, generates more nicked DNA than wild-type topo I in the presence of Genz. Based on a previous model in which the stable interaction of nitidine derivatives, having structural similarity to Genz, with arginine 364 and asparagine 722 of topo I inhibits the religation step, the authors have performed mutagenesis studies at these positions. The results allowed them to present a model explaining why the

N722S mutation make topo I resistant to camptothecin but not to Genz. Overall the results presented in this paper are clear and they support very well the conclusions. Furthermore, the results of experiments related to asparagine 722 residue may help to design better camptothecin-like topo I inhibitors.

Minor comment:

In the siRNA experiment in Fig. 2, a stronger Broken DNA signal is seen for Genz than for camptothecin in the absence of topo I. Does it means that Genz is less specific to topo I than camptothecin?

Reactions to reviewers' comments.

Reviewer #1 (Remarks to the Author):

In this manuscript, the authors analyze the mechanisms of action of Genz-644282, a non-camptothecin inhibitor of topoisomerase I (TOP1). In particular, they show that the cytotoxic activity of Genz-644282 is mainly related to the induction of replication-dependent DNA double-strand breaks (DSBs), a mechanism of action similar to that of camptothecin. They also show that, unlike camptothecin, Genz-644282 is still able to induce TOP1 trapping and cytotoxicity when TOP1 carries the N722S mutation. The experiments are generally sound and the conclusions are supported by the data. However, the main findings of this manuscript have been previously reported, including that Genz-644282 induces DSBs mostly in S-phase cells and overcomes camptothecin resistance in cells carrying the TOP1 N722S mutation (Sooryakumar et al, Mol Cancer Ther 2011). Therefore, this manuscript is rather incremental/complementary and does not significantly advance our knowledge of the mechanism of action of Genz-644282.

First of all, we thank all reviewers to give us many good suggestions. All comments were quite reasonable and contributed to improve our revised manuscript.

Minor points:

Introduction, line 48: It is now established that camptothecin also induces DSBs outside of S phase by a transcription-dependent mechanism, and that camptothecin is also cytotoxic in non-proliferating cells. This should be mentioned.

We add the sentence below in 'Introduction' in Line 46-37, Page 3.

'CPT also induces DSBs outside of S phase by a transcription-dependent mechanism, and that CPT is also cytotoxic in non-proliferating cells⁸.'

Drug concentration and treatment time are sometimes missing from figure legends.

We add concentration and treatment time in Figure legends: Line 640-656, Page 25.

Figure 1. Analysis of the cytotoxic effects of CPT and Genz. **a** Chemical structures of

camptothecin (CPT) and Genz-644282 (Genz). **b** Immunofluorescence analysis of the formation of EdU and γ -H2AX foci. Cells were pulse-labelled with 10 μ M EdU for 15 min. Subsequently, these pulse-labelled cells were treated with 1 μ M CPT or 1 μ M Genz, and the formation of EdU and γ -H2AX foci was analysed. **c** PFGE analysis of DSB accumulation after 24 h-treatment with a DNA replication inhibitor, aphidicolin (10 μ M), combined with 1 μ M CPT and 1 μ M Genz, and PFGE analysis of DSB accumulation after 24 h-treatment with an apoptosis inhibitor, Z-VAD-FMK (10 μ M), combined with 1 μ M CPT and 1 μ M Genz. **d** Quantification of CPT- and Genz-induced 'broken DNA', which represents DSB formation due to DNA stress. The data are presented as the percentages of the amount of broken DNA relative to total DNA (intact DNA + broken DNA + chromosome fragmentation in apoptosis). **e** Quantification of 'chromosome fragmentation in apoptosis' induced by treatments with CPT and Genz. The data are presented as the percentages of the amount of chromosome fragmentation in apoptosis per total DNA (intact DNA + broken DNA + chromosome fragmentation in apoptosis). The means and standard deviations were determined from four independent experiments. **f** Survival curve of SV40-transformed human fibroblasts, MRC5 against CPT. Indicated concentrations of CPT were applied for 24 h. **g** Survival curve of CPT on SV40-transformed MRC5 against Genz. Indicated concentrations of CPT was applied for 24 h.

Reviewer #2 (Remarks to the Author):

This manuscript by Nishida et al. focuses on Genz, a non-camptothecin inhibitor of human TOP1, that may overcome disadvantages of camptothecin for cancer therapy. The experiments are to provide information on the mechanism of action of Genz, and to demonstrate the effectiveness of Genz against cell lines with camptothecin (CPT) resistant mutation at residue 722 of TOP1. The experimental results are in general of high quality. Certain properties of camptothecin, including induction of gamma-H2AX followed by 53BP1, replication-dependence for generating DSB and cytotoxicity, could be observed for Genz. It is interesting that the results here showed that while TOP1 with N722S and N722A mutation are resistant to CPT for inhibition of DNA religation, increased TOP1 covalent complex was observed from Genz treatment of the mutant N722S and N722A TOP1. The manuscript also showed that Genz had good cytotoxicity against the CPT-resistant cell line CCRF-CEM with the N722S mutation in

TOP1. These results should be of interest for the potential application of Genz in cancer therapy. The results are also relevant for the understanding of human TOP1 interactions with inhibitors that may affect its cleavage-religation of DNA and the accumulation of the TOP1 covalent complex. However, there are significant weaknesses in the manuscript that need to be addressed.

1. According to 2000 review of camptothecin mechanism of action by Leroy Liu and others <https://pubmed.ncbi.nlm.nih.gov/11193884/>: “The primary mechanism of cell killing by CPT is S-phase-specific killing through potentially lethal collisions between advancing replication forks and topo-I cleavable complexes. Collisions with the transcription machinery have also been shown to trigger the formation of long-lived covalent topo-I DNA complexes, which contribute to CPT cytotoxicity.” The stabilization of the covalent TOP1-DNA covalent complex (TOP1cc) and associated DNA breaks may be more important than the defect in the relaxation of topological stress in front of the replication fork. It is puzzling that the percent of broken DNA shown from Genz treatment in Figure 2 is much lower than the percent of broken DNA shown from CPT treatment. Yet Genz is 10-fold more potent than CPT for decreasing cell survival (Figure 1 f-g). The treatment of TOP1 siRNA produced a much greater effect on TOP1 DNA breaks from CPT when compared to Genz (Figure 1c). These results suggest that there may be additional mechanism of action for Genz. Such possibility should be considered when stating that “the mechanism of action of Genz at the cellular level is the same as that of CPT” (line 258).

This point is quite important. To address this, we have done the time-course experiment and found that two distinct peaks of broken DNA were observed after treatment with Genz. The first peak appeared around 4 h and the second one appeared between 24 and 36 h (Fig. 2a, b). This result indicates that there are two distinct mechanisms of Genz-induced DSB formation. Similar to CPT, chromosome fragmentation in apoptosis gradually increased until 48 h after treatment with Genz (Fig. 2a, b). Important finding is that the accumulation of DSBs induced by treatment with Genz for 4 h was not reduced by MUS81 depletion (Fig. 2d, e). This indicated that Genz-induced DSB formation in the early response occurs without the action of MUS81–EME1/2.

We add these results in Figure 2, which described in Line 129-138, Page 8, and Line 152-158, Page 9, and discussed possible models in discussion, Line 302-314, Page 12.

Line 129-138, Page 8

Next, we performed time course experiments. After treatment with CPT and Genz, the

accumulation of broken DNA and chromosome fragmentation in apoptosis were assessed by PFGE. After treatment with CPT, broken DNA was detected from 16 to 48 h. The highest accumulation of broken DNA was observed at 24 h. Chromosome fragmentation in apoptosis gradually increased until 48 h (Fig. 2a, b). In contrast, two distinct peaks of broken DNA were observed after treatment with Genz. The first peak appeared around 4 h and the second one appeared between 24 and 36 h (Fig. 2a, b). This result indicates that there are two distinct mechanisms of Genz-induced DSB formation. Similar to CPT, chromosome fragmentation in apoptosis gradually increased until 48 h after treatment with Genz (Fig. 2a, b). This result also supports the idea that cells treated with Genz accumulate DSBs, which triggers apoptosis.

Line 152-158, Page 9

However, the accumulation of DSBs induced by treatment with Genz for 4 h was not reduced by MUS81 depletion (Fig. 2d, e). This indicated that Genz-induced DSB formation in the early response occurs without the action of MUS81–EME1/2.

On the basis of these results, we suggest that Genz treatment initially induces TOP1-dependent but MUS81–EME1/2-independent DSB formation in the early response, and a majority of stalled forks are cleaved by MUS81–EME1/2 structure-specific endonuclease in the late response, thus inducing DSBs around DNA replication sites.

Line 302-314, Page 12.

In this study, we found two different mechanisms of Genz-induced DSB formation. One was the early response, which occurred within 8 h, requiring TOP1 function but not MUS81–EME1/2 function, while the other was the late response, which appeared later than at 16 h of incubation, requiring both TOP1 and MUS81–EME1/2 functions (Fig. 2c, d). In this context, a question arises about the mechanism through which DSB formation occurs after treatment with Genz. Because Genz could stabilise the TOP1–DNA complex (Fig. 3, 7, 8), one possibility is that a single-strand break, formed as a result of stabilisation of the TOP1–DNA complex by treatment with Genz, might be converted into a DSB through progression of the DNA replication fork. The majority of Genz-induced DSB formation in the early response likely occurs through this mechanism (Fig. 9a). Meanwhile, Genz-induced DSB formation in the late response required the action of MUS81–EME1/2 function (Fig. 2c, d). In this case, stalled replication forks were initially induced by the inhibition of TOP1 function by treatment with Genz, and then some of the stalled replication forks were cleaved by a structure-specific endonuclease, MUS81–EME1/2,

which results in the formation of DSBs (Fig. 9a).

2. Since it is possible that there is additional target for Genz that is independent of TOP1, selection for Genz-resistance and characterization of the associated mutations are needed to confirm that TOP1 is the major cellular target for Genz. The characterization of Genz resistance would also help establish if Genz resistance arises at similar rate as CPT resistance and if the Genz resistance affects the proliferation rate of the cancer cells. Such information is needed to evaluate Genz as alternative treatment option in the clinic.

Following the editor's suggestions, we have characterized the D533G- and N722S-mutated cells. Because D533G mutation causes Genz-resistance, cells with D533G mutation in TOP1 gene should not induce DSB formation after treatment with Genz. We have added new results in Figure 4 and 5. As far as Genz-Induced DSB formation is concerned, we could not find any other target.

We added the new sentences, in Line 197- 202, Page8, and Line 208-220, Page 9.

Line 197- 202, Page8

To investigate whether Genz-induced DSB formation is correlated to its cytotoxicity, the accumulation of DSBs after treatment with CPT and Genz was analysed by PFGE using the CPT-resistant but Genz-sensitive cell line, CEM/C2. CPT-induced DSB formation was not observed but Genz-induced DSB formation was detected in CEM/C2 cells (Fig. 4c, d). Meanwhile, the formation of both CPT- and Genz-induced DSBs was observed in CCRF-CEM cells (Fig. 4c, d).

Line 208-220, Page 9

Next, the accumulation of DSBs after treatment with CPT and Genz was analysed by PFGE using the CPT- and Genz-resistant cell line, CPT-K5. Because the survival rates of CPT-K5 after treatment with 0.1 μ M and 1 μ M Genz were 100% and 27%, and those of RPMI-8402 were 3.6% and 3.8%, respectively (Fig. 5a), RPMI-8402 and CPT-K5 cells were treated with 0.1 μ M and 1 μ M Genz, and the accumulation of DSBs was investigated by PFGE. DSB formation was observed in RPMI-8402 cells both after treatment with both 0.1 μ M and 1 μ M Genz (Fig. 5b, e). Meanwhile, no DSB accumulation was observed after treatment with 0.1 μ M Genz, and slightly but significantly higher accumulation of DSBs was observed after treatment with 1 μ M Genz in CPT-K5 cells (Fig. 5b, e). In contrast, CPT-induced DSB formation was observed in RPMI-8402 cells

but not in CPT-K5 cells (Fig. 5d, f). Because the survival rate of RPMI-8402 after treatment with 1 μ M CPT was 24 % and that of CPT-K5 was 99%, the accumulation of DSBs after treatments with CPT was correlated to its cytotoxicity (Fig. 5b, d, f). These results suggested that DSB formation induced by the treatment with CPT and Genz must be a major cause of cell death.

3. I cannot find the drug concentrations used to generate the data in Figure 1b-e. These drug concentrations should have been included in the Figure 1 legend.

This point is the same as the one reviewer 1 has been pointed out. We have added the drug concentrations in the Figure 1 legend (See above).

4. There are a number of instances where use of English needs to be corrected. For example:

i. line 38, “cutting of the phosphate group” – should be linking to the phosphate group

Fixed (Line 37).

ii. line 78, “found for removing another CPT-resistant cell line”

Fixed (Line 78).

iii. line 85, “Genz acts to CPT-resistant TOP1”

Fixed (Line 84-85).

iv: line 105, “it was suggested that stalled DNA..”

Fixed (Line 104).

v: line 228, “reduced by one-three hundredth of that”, should be “reduced to one-three hundredth of that”

Fixed (Line 273).

5. The sources of chemicals (Genz and CPT) should be described under Methods.

We added the sentence below in Methods in Line 375-378, Page 14.

Camptothecin (CAS No. 7689-03-4) was purchased from FujiFilm co ltd., and Genz-644282 (CAS No. 529488-28-6) was purchased from Selleck Chemicals co ltd and MedChemExpress (MCE) co ltd.

Reviewer #3 (Remarks to the Author):

In this paper, the authors presented their results about the non-camptothecin topo I inhibitor, Genz-644282. They have compared the effects of camptothecin and Genz in vitro on purified topo I as well as on 14 purified topo I camptothecin-resistant mutants. They have also studied in detail the mechanism of Genz cytotoxicity in cells expressing wild type topo I. Genz toxicity on 14 cell lines each expressing a different camptothecin-resistant mutant has also been evaluated. They found that the mechanism of cytotoxicity of Genz was very similar to that of camptothecin i.e., stalling of replication forks through topo I inhibition, followed by cleavage of the forks by MUS81-EME1/2 structure-specific endonuclease inducing DSBs. This is followed by apoptosis. The most interesting finding (novelty) of the paper stem from the observation that the only camptothecin-resistant topo I mutant, N722S, that is still sensitive to Genz both in vitro and in vivo, generates more nicked DNA than wild-type topo I in the presence of Genz. Based on a previous model in which the stable interaction of nitidine derivatives, having structural similarity to Genz, with arginine 364 and asparagine 722 of topo I inhibits the religation step, the authors have performed mutagenesis studies at these positions. The results allowed them to present a model explaining why the N722S mutation make topo I resistant to camptothecin but not to Genz. Overall the results presented in this paper are clear and they support very well the conclusions. Furthermore, the results of experiments related to asparagine 722 residue may help to design better camptothecin-like topo I inhibitors.

Minor comment:

In the siRNA experiment in Fig. 2, a stronger Broken DNA signal is seen for Genz than for camptothecin in the absence of topo I. Does it mean that Genz is less specific to topo I than camptothecin?

This point is the same as the one reviewer 2 has been pointed out. We add these results in Figure 2, and described in Line 129-138, Page 8, and Line 152-158, Page 9 (See above).

Reviewers' comments:

Reviewer #2 (Remarks to the Author):

The revision has mostly addressed my previous comments. However, the following still requires the authors' attention:

1. The authors should check that in Figure 5e, the broken DNA generated by 1 μ M Genz at 24 h in CPT-K5 cells is indeed significantly greater than the amount of broken DNA at 0 h ($p < 0.05$). It does not look convincing based on the visual inspection.
2. The legend of Figure 6 does not agree with the labeling of the panels. Typos need to be corrected.
3. Figure 9 and Supplementary Figure S4: N722 should be labeled as Asn722, not Apn722.

Reactions to reviewer's comments.

Reviewers' comments:

Reviewer #2 (Remarks to the Author):

The revision has mostly addressed my previous comments. However, the following still requires the authors' attention:

1. The authors should check that in Figure 5e, the broken DNA generated by 1 μ M Genz at 24 h in CPT-K5 cells is indeed significantly greater than the amount of broken DNA at 0 h ($p < 0.05$). It does not look convincing based on the visual inspection.

We have done 4 independent experiments. Percentages of broken DNA per total DNA generated by 1 μ M Genz at 0h in CPT-K5 cells were 11.7%, 4.68%, 4.2%, and 3.79%, respectively. Therefore, the mean was 6.1%, and its SD was 1.9%. On the other hand, percentages of broken DNA per total DNA generated by 1 μ M Genz at 24h in CPT-K5 cells were 11.8%, 5.05%, 4.84%, and 4.71%, respectively. The mean was 6.6%, and its SD was 3.5%. Using these data, we have performed one-tailed t-test, and its result was $p=0.0318$. However, as the reviewer mentioned, the difference between these, 6.1% VS 6.6%, was too small, it may be statistical analysis was meaningless. Therefore, we omit p value of this. Even if we omit this, it does not change our conclusion.

2. The legend of Figure 6 does not agree with the labeling of the panels. Typos need to be corrected.

Fixed.

3. Figure 9 and Supplementary Figure S4: N722 should be labeled as Asn722, not Apn722.

Fixed.

Reviewers' comments:

Reviewer #1 (Remarks to the Author):

In this manuscript, the authors analyze the mechanisms of action of Genz-644282, a non-camptothecin inhibitor of topoisomerase I (TOP1). In particular, they show that the cytotoxic activity of Genz-644282 is mainly related to the induction of replication-dependent DNA double-strand breaks (DSBs), a mechanism of action similar to that of camptothecin. They also show that, unlike camptothecin, Genz-644282 is still able to induce TOP1 trapping and cytotoxicity when TOP1 carries the N722S mutation. The experiments are generally sound and the conclusions are supported by the data. However, the main findings of this manuscript have been previously reported, including that Genz-644282 induces DSBs mostly in S-phase cells and overcomes camptothecin resistance in cells carrying the TOP1 N722S mutation (Sooryakumar et al, Mol Cancer Ther 2011). Therefore, this manuscript is rather incremental/complementary and does not significantly advance our knowledge of the mechanism of action of Genz-644282.

Minor points:

Introduction, line 48: It is now established that camptothecin also induces DSBs outside of S phase by a transcription-dependent mechanism, and that camptothecin is also cytotoxic in non-proliferating cells. This should be mentioned.

Drug concentration and treatment time are sometimes missing from figure legends.

Reviewer #2 (Remarks to the Author):

This manuscript by Nishida et al. focuses on Genz, a non-camptothecin inhibitor of human TOP1, that may overcome disadvantages of camptothecin for cancer therapy. The experiments are to provide information on the mechanism of action of Genz, and to demonstrate the effectiveness of Genz against cell lines with camptothecin (CPT) resistant mutation at residue

722 of TOP1. The experimental results are in general of high quality. Certain properties of camptothecin, including induction of gamma-H2AX followed by 53BP1, replication-dependence for generating DSB and cytotoxicity, could be observed for Genz. It is interesting that the results here showed that while TOP1 with N722S and N722A mutation are resistant to CPT for inhibition of DNA religation, increased TOP1 covalent complex was observed from Genz treatment of the mutant N722S and N722A TOP1. The manuscript also showed that Genz had good cytotoxicity against the CPT-resistant cell line CCRF-CEM with the N722S mutation in TOP1. These results should be of interest for the potential application of Genz in cancer therapy. The results are also relevant for the understanding of human TOP1 interactions with inhibitors that may affect its cleavage-religation of DNA and the accumulation of the TOP1 covalent complex. However, there are significant weaknesses in the manuscript that need to be addressed.

1. According to 2000 review of camptothecin mechanism of action by Leroy Liu and others <https://pubmed.ncbi.nlm.nih.gov/11193884/>: "The primary mechanism of cell killing by CPT is S-phase-specific killing through potentially lethal collisions between advancing replication forks and topo-I cleavable complexes. Collisions with the transcription machinery have also been shown to trigger the formation of long-lived covalent topo-I DNA complexes, which contribute to CPT cytotoxicity." The stabilization of the covalent TOP1-DNA covalent complex (TOP1cc) and associated DNA breaks may be more important than the defect in the relaxation of topological stress in front of the replication fork. It is puzzling that the percent of broken DNA shown from Genz treatment in Figure 2 is much lower than the percent of broken DNA shown from CPT treatment. Yet Genz is 10-fold more potent than CPT for decreasing cell survival (Figure 1 f-g). The treatment of TOP1 siRNA produced a much greater effect on TOP1 DNA breaks from CPT when compared to Genz (Figure 1c). These results suggest that there may be additional mechanism of action for Genz. Such possibility should be considered when stating that "the mechanism of action of Genz at the cellular level is the same as that of CPT" (line 258).

2. Since it is possible that there is additional target for Genz that is independent of TOP1, selection for Genz-resistance and characterization of the associated mutations are needed to confirm that TOP1 is the major cellular target for Genz. The characterization of Genz resistance would also help establish if Genz resistance arises at similar rate as CPT resistance and if the Genz resistance affects the proliferation rate of the cancer cells. Such information is needed to

evaluate Genz as alternative treatment option in the clinic.

3. I cannot find the drug concentrations used to generate the data in Figure 1b-e. These drug concentrations should have been included in the Figure 1 legend.

4. There are a number of instances where use of English needs to be corrected. For example:

i. line 38, “cutting of the phosphate group” – should be linking to the phosphate group

ii. line 78, “found for removing another CPT-resistant cell line”

iii. line 85, “Genz acts to CPT-resistant TOP1”

iv: line 105, “it was suggested that stalled DNA..”

v: line 228, “reduced by one-three hundredth of that”, should be “reduced to one-three hundredth of that”

5. The sources of chemicals (Genz and CPT) should be described under Methods.

Reviewer #3 (Remarks to the Author):

In this paper, the authors presented their results about the non-camptothecin topo I inhibitor, Genz-644282. They have compared the effects of camptothecin and Genz in vitro on purified topo I as well as on 14 purified topo I camptothecin-resistant mutants. They have also studied in detail the mechanism of Genz cytotoxicity in cells expressing wild type topo I. Genz toxicity on 14 cell lines each expressing a different camptothecin-resistant mutant has also been evaluated. They found that the mechanism of cytotoxicity of Genz was very similar to that of camptothecin i.e., stalling of replication forks through topo I inhibition, followed by cleavage of the forks by MUS81-EME1/2 structure-specific endonuclease inducing DSBs. This is followed by apoptosis. The most interesting finding (novelty) of the paper stem from the observation that the only camptothecin-resistant topo I mutant, N722S, that is still sensitive to Genz both in vitro and in vivo, generates more nicked DNA than wild-type topo I in the presence of Genz. Based on a previous model in which the stable interaction of nitidine derivatives, having structural similarity to Genz, with arginine 364 and asparagine 722 of topo I inhibits the religation step, the authors have performed mutagenesis studies at these positions. The results allowed them to present a model explaining why the

N722S mutation make topo I resistant to camptothecin but not to Genz. Overall the results presented in this paper are clear and they support very well the conclusions. Furthermore, the results of experiments related to asparagine 722 residue may help to design better camptothecin-like topo I inhibitors.

Minor comment:

In the siRNA experiment in Fig. 2, a stronger Broken DNA signal is seen for Genz than for camptothecin in the absence of topo I. Does it means that Genz is less specific to topo I than camptothecin?

Reactions to reviewers' comments.

Reviewer #1 (Remarks to the Author):

In this manuscript, the authors analyze the mechanisms of action of Genz-644282, a non-camptothecin inhibitor of topoisomerase I (TOP1). In particular, they show that the cytotoxic activity of Genz-644282 is mainly related to the induction of replication-dependent DNA double-strand breaks (DSBs), a mechanism of action similar to that of camptothecin. They also show that, unlike camptothecin, Genz-644282 is still able to induce TOP1 trapping and cytotoxicity when TOP1 carries the N722S mutation. The experiments are generally sound and the conclusions are supported by the data. However, the main findings of this manuscript have been previously reported, including that Genz-644282 induces DSBs mostly in S-phase cells and overcomes camptothecin resistance in cells carrying the TOP1 N722S mutation (Sooryakumar et al, Mol Cancer Ther 2011). Therefore, this manuscript is rather incremental/complementary and does not significantly advance our knowledge of the mechanism of action of Genz-644282.

First of all, we thank all reviewers to give us many good suggestions. All comments were quite reasonable and contributed to improve our revised manuscript.

Minor points:

Introduction, line 48: It is now established that camptothecin also induces DSBs outside of S phase by a transcription-dependent mechanism, and that camptothecin is also cytotoxic in non-proliferating cells. This should be mentioned.

We add the sentence below in 'Introduction' in Line 46-37, Page 3.

'CPT also induces DSBs outside of S phase by a transcription-dependent mechanism, and that CPT is also cytotoxic in non-proliferating cells⁸.'

Drug concentration and treatment time are sometimes missing from figure legends.

We add concentration and treatment time in Figure legends: Line 640-656, Page 25.

Figure 1. Analysis of the cytotoxic effects of CPT and Genz. **a** Chemical structures of

camptothecin (CPT) and Genz-644282 (Genz). **b** Immunofluorescence analysis of the formation of EdU and γ -H2AX foci. Cells were pulse-labelled with 10 μ M EdU for 15 min. Subsequently, these pulse-labelled cells were treated with 1 μ M CPT or 1 μ M Genz, and the formation of EdU and γ -H2AX foci was analysed. **c** PFGE analysis of DSB accumulation after 24 h-treatment with a DNA replication inhibitor, aphidicolin (10 μ M), combined with 1 μ M CPT and 1 μ M Genz, and PFGE analysis of DSB accumulation after 24 h-treatment with an apoptosis inhibitor, Z-VAD-FMK (10 μ M), combined with 1 μ M CPT and 1 μ M Genz. **d** Quantification of CPT- and Genz-induced 'broken DNA', which represents DSB formation due to DNA stress. The data are presented as the percentages of the amount of broken DNA relative to total DNA (intact DNA + broken DNA + chromosome fragmentation in apoptosis). **e** Quantification of 'chromosome fragmentation in apoptosis' induced by treatments with CPT and Genz. The data are presented as the percentages of the amount of chromosome fragmentation in apoptosis per total DNA (intact DNA + broken DNA + chromosome fragmentation in apoptosis). The means and standard deviations were determined from four independent experiments. **f** Survival curve of SV40-transformed human fibroblasts, MRC5 against CPT. Indicated concentrations of CPT were applied for 24 h. **g** Survival curve of CPT on SV40-transformed MRC5 against Genz. Indicated concentrations of CPT was applied for 24 h.

Reviewer #2 (Remarks to the Author):

This manuscript by Nishida et al. focuses on Genz, a non-camptothecin inhibitor of human TOP1, that may overcome disadvantages of camptothecin for cancer therapy. The experiments are to provide information on the mechanism of action of Genz, and to demonstrate the effectiveness of Genz against cell lines with camptothecin (CPT) resistant mutation at residue 722 of TOP1. The experimental results are in general of high quality. Certain properties of camptothecin, including induction of gamma-H2AX followed by 53BP1, replication-dependence for generating DSB and cytotoxicity, could be observed for Genz. It is interesting that the results here showed that while TOP1 with N722S and N722A mutation are resistant to CPT for inhibition of DNA religation, increased TOP1 covalent complex was observed from Genz treatment of the mutant N722S and N722A TOP1. The manuscript also showed that Genz had good cytotoxicity against the CPT-resistant cell line CCRF-CEM with the N722S mutation in

TOP1. These results should be of interest for the potential application of Genz in cancer therapy. The results are also relevant for the understanding of human TOP1 interactions with inhibitors that may affect its cleavage-religation of DNA and the accumulation of the TOP1 covalent complex. However, there are significant weaknesses in the manuscript that need to be addressed.

1. According to 2000 review of camptothecin mechanism of action by Leroy Liu and others <https://pubmed.ncbi.nlm.nih.gov/11193884/>: “The primary mechanism of cell killing by CPT is S-phase-specific killing through potentially lethal collisions between advancing replication forks and topo-I cleavable complexes. Collisions with the transcription machinery have also been shown to trigger the formation of long-lived covalent topo-I DNA complexes, which contribute to CPT cytotoxicity.” The stabilization of the covalent TOP1-DNA covalent complex (TOP1cc) and associated DNA breaks may be more important than the defect in the relaxation of topological stress in front of the replication fork. It is puzzling that the percent of broken DNA shown from Genz treatment in Figure 2 is much lower than the percent of broken DNA shown from CPT treatment. Yet Genz is 10-fold more potent than CPT for decreasing cell survival (Figure 1 f-g). The treatment of TOP1 siRNA produced a much greater effect on TOP1 DNA breaks from CPT when compared to Genz (Figure 1c). These results suggest that there may be additional mechanism of action for Genz. Such possibility should be considered when stating that “the mechanism of action of Genz at the cellular level is the same as that of CPT” (line 258).

This point is quite important. To address this, we have done the time-course experiment and found that two distinct peaks of broken DNA were observed after treatment with Genz. The first peak appeared around 4 h and the second one appeared between 24 and 36 h (Fig. 2a, b). This result indicates that there are two distinct mechanisms of Genz-induced DSB formation. Similar to CPT, chromosome fragmentation in apoptosis gradually increased until 48 h after treatment with Genz (Fig. 2a, b). Important finding is that the accumulation of DSBs induced by treatment with Genz for 4 h was not reduced by MUS81 depletion (Fig. 2d, e). This indicated that Genz-induced DSB formation in the early response occurs without the action of MUS81–EME1/2.

We add these results in Figure 2, which described in Line 129-138, Page 8, and Line 152-158, Page 9, and discussed possible models in discussion, Line 302-314, Page 12.

Line 129-138, Page 8

Next, we performed time course experiments. After treatment with CPT and Genz, the

accumulation of broken DNA and chromosome fragmentation in apoptosis were assessed by PFGE. After treatment with CPT, broken DNA was detected from 16 to 48 h. The highest accumulation of broken DNA was observed at 24 h. Chromosome fragmentation in apoptosis gradually increased until 48 h (Fig. 2a, b). In contrast, two distinct peaks of broken DNA were observed after treatment with Genz. The first peak appeared around 4 h and the second one appeared between 24 and 36 h (Fig. 2a, b). This result indicates that there are two distinct mechanisms of Genz-induced DSB formation. Similar to CPT, chromosome fragmentation in apoptosis gradually increased until 48 h after treatment with Genz (Fig. 2a, b). This result also supports the idea that cells treated with Genz accumulate DSBs, which triggers apoptosis.

Line 152-158, Page 9

However, the accumulation of DSBs induced by treatment with Genz for 4 h was not reduced by MUS81 depletion (Fig. 2d, e). This indicated that Genz-induced DSB formation in the early response occurs without the action of MUS81–EME1/2.

On the basis of these results, we suggest that Genz treatment initially induces TOP1-dependent but MUS81–EME1/2-independent DSB formation in the early response, and a majority of stalled forks are cleaved by MUS81–EME1/2 structure-specific endonuclease in the late response, thus inducing DSBs around DNA replication sites.

Line 302-314, Page 12.

In this study, we found two different mechanisms of Genz-induced DSB formation. One was the early response, which occurred within 8 h, requiring TOP1 function but not MUS81–EME1/2 function, while the other was the late response, which appeared later than at 16 h of incubation, requiring both TOP1 and MUS81–EME1/2 functions (Fig. 2c, d). In this context, a question arises about the mechanism through which DSB formation occurs after treatment with Genz. Because Genz could stabilise the TOP1–DNA complex (Fig. 3, 7, 8), one possibility is that a single-strand break, formed as a result of stabilisation of the TOP1–DNA complex by treatment with Genz, might be converted into a DSB through progression of the DNA replication fork. The majority of Genz-induced DSB formation in the early response likely occurs through this mechanism (Fig. 9a). Meanwhile, Genz-induced DSB formation in the late response required the action of MUS81–EME1/2 function (Fig. 2c, d). In this case, stalled replication forks were initially induced by the inhibition of TOP1 function by treatment with Genz, and then some of the stalled replication forks were cleaved by a structure-specific endonuclease, MUS81–EME1/2,

which results in the formation of DSBs (Fig. 9a).

2. Since it is possible that there is additional target for Genz that is independent of TOP1, selection for Genz-resistance and characterization of the associated mutations are needed to confirm that TOP1 is the major cellular target for Genz. The characterization of Genz resistance would also help establish if Genz resistance arises at similar rate as CPT resistance and if the Genz resistance affects the proliferation rate of the cancer cells. Such information is needed to evaluate Genz as alternative treatment option in the clinic.

Following the editor's suggestions, we have characterized the D533G- and N722S-mutated cells. Because D533G mutation causes Genz-resistance, cells with D533G mutation in TOP1 gene should not induce DSB formation after treatment with Genz. We have added new results in Figure 4 and 5. As far as Genz-Induced DSB formation is concerned, we could not find any other target.

We added the new sentences, in Line 197- 202, Page8, and Line 208-220, Page 9.

Line 197- 202, Page8

To investigate whether Genz-induced DSB formation is correlated to its cytotoxicity, the accumulation of DSBs after treatment with CPT and Genz was analysed by PFGE using the CPT-resistant but Genz-sensitive cell line, CEM/C2. CPT-induced DSB formation was not observed but Genz-induced DSB formation was detected in CEM/C2 cells (Fig. 4c, d). Meanwhile, the formation of both CPT- and Genz-induced DSBs was observed in CCRF-CEM cells (Fig. 4c, d).

Line 208-220, Page 9

Next, the accumulation of DSBs after treatment with CPT and Genz was analysed by PFGE using the CPT- and Genz-resistant cell line, CPT-K5. Because the survival rates of CPT-K5 after treatment with 0.1 μ M and 1 μ M Genz were 100% and 27%, and those of RPMI-8402 were 3.6% and 3.8%, respectively (Fig. 5a), RPMI-8402 and CPT-K5 cells were treated with 0.1 μ M and 1 μ M Genz, and the accumulation of DSBs was investigated by PFGE. DSB formation was observed in RPMI-8402 cells both after treatment with both 0.1 μ M and 1 μ M Genz (Fig. 5b, e). Meanwhile, no DSB accumulation was observed after treatment with 0.1 μ M Genz, and slightly but significantly higher accumulation of DSBs was observed after treatment with 1 μ M Genz in CPT-K5 cells (Fig. 5b, e). In contrast, CPT-induced DSB formation was observed in RPMI-8402 cells

but not in CPT-K5 cells (Fig. 5d, f). Because the survival rate of RPMI-8402 after treatment with 1 μ M CPT was 24 % and that of CPT-K5 was 99%, the accumulation of DSBs after treatments with CPT was correlated to its cytotoxicity (Fig. 5b, d, f). These results suggested that DSB formation induced by the treatment with CPT and Genz must be a major cause of cell death.

3. I cannot find the drug concentrations used to generate the data in Figure 1b-e. These drug concentrations should have been included in the Figure 1 legend.

This point is the same as the one reviewer 1 has been pointed out. We have added the drug concentrations in the Figure 1 legend (See above).

4. There are a number of instances where use of English needs to be corrected. For example:

i. line 38, “cutting of the phosphate group” – should be linking to the phosphate group

Fixed (Line 37).

ii. line 78, “found for removing another CPT-resistant cell line”

Fixed (Line 78).

iii. line 85, “Genz acts to CPT-resistant TOP1”

Fixed (Line 84-85).

iv: line 105, “it was suggested that stalled DNA..”

Fixed (Line 104).

v: line 228, “reduced by one-three hundredth of that”, should be “reduced to one-three hundredth of that”

Fixed (Line 273).

5. The sources of chemicals (Genz and CPT) should be described under Methods.

We added the sentence below in Methods in Line 375-378, Page 14.

Camptothecin (CAS No. 7689-03-4) was purchased from FujiFilm co ltd., and Genz-644282 (CAS No. 529488-28-6) was purchased from Selleck Chemicals co ltd and MedChemExpress (MCE) co ltd.

Reviewer #3 (Remarks to the Author):

In this paper, the authors presented their results about the non-camptothecin topo I inhibitor, Genz-644282. They have compared the effects of camptothecin and Genz in vitro on purified topo I as well as on 14 purified topo I camptothecin-resistant mutants. They have also studied in detail the mechanism of Genz cytotoxicity in cells expressing wild type topo I. Genz toxicity on 14 cell lines each expressing a different camptothecin-resistant mutant has also been evaluated. They found that the mechanism of cytotoxicity of Genz was very similar to that of camptothecin i.e., stalling of replication forks through topo I inhibition, followed by cleavage of the forks by MUS81-EME1/2 structure-specific endonuclease inducing DSBs. This is followed by apoptosis. The most interesting finding (novelty) of the paper stems from the observation that the only camptothecin-resistant topo I mutant, N722S, that is still sensitive to Genz both in vitro and in vivo, generates more nicked DNA than wild-type topo I in the presence of Genz. Based on a previous model in which the stable interaction of nitidine derivatives, having structural similarity to Genz, with arginine 364 and asparagine 722 of topo I inhibits the religation step, the authors have performed mutagenesis studies at these positions. The results allowed them to present a model explaining why the N722S mutation makes topo I resistant to camptothecin but not to Genz. Overall the results presented in this paper are clear and they support very well the conclusions. Furthermore, the results of experiments related to asparagine 722 residue may help to design better camptothecin-like topo I inhibitors.

Minor comment:

In the siRNA experiment in Fig. 2, a stronger Broken DNA signal is seen for Genz than for camptothecin in the absence of topo I. Does it mean that Genz is less specific to topo I than camptothecin?

This point is the same as the one reviewer 2 has been pointed out. We add these results in Figure 2, and described in Line 129-138, Page 8, and Line 152-158, Page 9 (See above).

Reactions to reviewer's comments.

Reviewers' comments:

Reviewer #2 (Remarks to the Author):

The revision has mostly addressed my previous comments. However, the following still requires the authors' attention:

1. The authors should check that in Figure 5e, the broken DNA generated by 1 μ M Genz at 24 h in CPT-K5 cells is indeed significantly greater than the amount of broken DNA at 0 h ($p < 0.05$). It does not look convincing based on the visual inspection.

We have done 4 independent experiments. Percentages of broken DNA per total DNA generated by 1 μ M Genz at 0h in CPT-K5 cells were 11.7%, 4.68%, 4.2%, and 3.79%, respectively. Therefore, the mean was 6.1%, and its SD was 1.9%. On the other hand, percentages of broken DNA per total DNA generated by 1 μ M Genz at 24h in CPT-K5 cells were 11.8%, 5.05%, 4.84%, and 4.71%, respectively. The mean was 6.6%, and its SD was 3.5%. Using these data, we have performed one-tailed t-test, and its result was $p=0.0318$. However, as the reviewer mentioned, the difference between these, 6.1% VS 6.6%, was too small, it may be statistical analysis was meaningless. Therefore, we omit p value of this. Even if we omit this, it does not change our conclusion.

2. The legend of Figure 6 does not agree with the labeling of the panels. Typos need to be corrected.

Fixed.

3. Figure 9 and Supplementary Figure S4: N722 should be labeled as Asn722, not Apn722.

Fixed.